# Citrullinated fibrinogen-SAAs complex causes vascular metastagenesis

Yibing Han[1,2], Takeshi Tomita [1,2], Masayoshi Kato [1,2], Norihiro Ashihara [1,2], Yumiko Higuchi[3,4], Hisanori Matoba[5], Weiyi Wang [1,2], Hikaru Hayashi[1,2], Yuji Itoh[6], Satoshi Takahashi[6], Hiroshi Kurita[7], Jun Nakayama[5], Nobuo Okumura [3,4] & Sachie Hiratsuka [1,2] ✉

Primary tumor cells metastasize to a distant preferred organ. However, the most decisive host factors that determine the precise locations of metastases in cancer patients remain unknown. We have demonstrated that post-translational citrullination of fibrinogen creates a metastatic niche in the vulnerable spots. Pulmonary endothelial cells mediate the citrullination of fibrinogen, changing its conformation, surface charge, and binding properties with serum amyloid A proteins (SAAs), to make it a host tissue-derived metastatic pathogen. The human-specific SAAs-citrullinated fibrinogen (CitFbg) complex recruits cancer cells to form a protein-metastatic cell aggregation in humanized *SAA* cluster mice. Furthermore, a CitFbg peptide works as a competitive inhibitor to block the homing of metastatic cells into the SAAs-CitFbg sites. The potential metastatic sites in the lungs of patients are clearly visualized by our specific antibody for CitFbg. Thus, CitFbg deposition displays metastatic risks for cancer patients, and the citrullinated peptide is a new type of metastasis inhibitor.

Metastasis is the last step in cancer progression, and anti-metastatic drugs have been developed to suppress this stage. Metastasis involves several steps that the primary tumor cells need to take, such as intravasation from the primary site, circulation in vessels, and attachment to the endothelial bed in a secondary organ, followed by extravasation into the tissues[1–4]. The pre-metastatic phase is considered as the host reaction, before the metastatic tumor cells physically appear in the secondary organ[5–7]. Lung metastasis in mouse model systems has been extensively investigated. This research data has demonstrated that regulation of stromal environments induces the settlement of circulating tumor cells[5,8–12]. Indeed, some pilot mouse studies were designed to decimate the pre-metastatic scaffold or to detect the pre-metastatic region[13,14]. This research would be very beneficial for cancer patients, as

this would alert them in the very early phase, known as the pre-metastatic phase, which would help to prevent initial metastasis and occlude the secondary metastasis[15], because of which, their prognosis would be far better than those requiring remedying of established metastases. The most critical point in this regard is, whether the pre-metastatic region can be detected in the clinical context, and if detected, can metastasis be prevented. Previously, we had discovered that fibrinogen depositions, discretely distributed in the pulmonary blood vessels, were sites with a high potential for metastasis, based on the findings in our mouse model system and specimens from cancer patients[16]. First, it was observed that the hyperpermeable regions with fibrinogen deposition attract circulating tumor cells in the lungs of pre-metastatic mice[16]. Secondly, when we used coagulation factor X

[1]Institute for Biomedical Sciences, Interdisciplinary Cluster for Cutting Edge Research, Shinshu University School of Medicine, Matsumoto, Japan. [2]Department of Biochemistry and Molecular Biology, Shinshu University School of Medicine, Matsumoto, Japan. [3]Department of Health and Medical Sciences, Graduate School of Medicine, Shinshu University, Matsumoto, Japan. [4]Department of Biomedical Laboratory Sciences, Shinshu University School of Medicine, Matsumoto, Japan. [5]Department of Molecular Pathology, Shinshu University School of Medicine, Matsumoto, Japan. [6]Institute of Multidisciplinary Research for Advanced Materials, Tohoku University, Sendai, Japan. [7]Department of Dentistry and Oral Surgery, Shinshu University School of Medicine, Matsumoto, Japan. ✉e-mail: hira@shinshu-u.ac.jp

(FX)-deficient mice, we found out that the culprit of this phenomenon was not fibrin, but fibrinogen. In this mouse model, it was observed that the coagulation cascade is interrupted before the fibrinogen-fibrin conversion. As a result, the fibrinogen level was higher in the FX-deficient mice as compared to wild-type mice and, the FX-deficient mice overpowered the hyperpermeable regions[17]. Thirdly, we identified the presumptive pre-metastatic regions with fibrinogen deposition in autopsy lung samples from cancer patients[16]. The regions were relatively large areas marked with pre-metastatic signature molecules and fibrinogen. Lung metastasis has been expected to occur in the alveolocapillary, which has a characteristically large surface area (-70 m²). We found that fibrinogen deposition occupied ~5% of the area[16] (3.5 m²). The pre-metastatic niche should be specified on the size of 3–5 cm² (-1 cm³ of tumor mass area) as the unfolding surface area of alveoli in cancer patients because the actual metastasis is initiated at very few sites.

Hence, we sought detailed information about fibrinogen, including its post-translational modifications. The human genome project has identified around 20,000 to 25,000 genes. In contrast, the transcriptome comprises 100,000 transcripts, and the proteome exceeds 1 million proteins[18]. Post-translational modifications (PTM) expand the chemical repertoire of the standard 20 amino acids, regulating confirmation, enzymatic activities, and stability of the proteins. However, PTMs can sometimes be harmful, and some abnormal modifications are involved in various human diseases, including cancer[19]. In this study, we have demonstrated that citrullination, one of the PTMs, recruits metastasis (The summary model in Fig. 1a). This modification is induced by the host tissues'-derived enzymatic ability via peptidyl arginine deiminases (PADs), a family of conserved enzymes distributed in various tissues, including the lungs[20,21]. It has been reported that lung tissue contains PAD2 and PAD4 isoforms along with monocytes/macrophages and neutrophils have active PADs on the cell surface and, occasionally, secretes them in connection with inflammation[22]. It is necessary to set up a prospective research to determine the citrullinated locations, that could enable the accurate prediction of metastatic sites. Our strategy in this study was to employ humanized mouse systems to discover causative molecules. The resulting molecules were tested in various clinical samples to determine if our theory could be applied to forecast metastasis with high accuracy.

Here, we show that citrullinated fibrinogen complexes with SAA are present in the pre-metastatic lung and they facilitate tumor metastasis by directly interacting with circulating tumor cells. We also demonstrate that autopsy samples show depositions of citrullinated fibrinogen in the lungs, although they have no lung metastasis. Our antibody recognizes citrullinated fibrinogen, but not unmodified fibrinogen, and humanized mouse metastasis assay clearly shows that synthetic citrullinated peptide prevents metastasis. These results indicate that citrullinated fibrinogen is a potential target to diagnose and prevent metastasis in the very early stages.

## Results

### Pre-metastatic-signature genes in cancer patients

We prepared serial sections of autopsied lung tissues obtained from noncancer and cancer patients to search for critical molecules in the fibrinogen deposition areas. It should be noted that none of the lung lobes of cancer patients had any signs of hemorrhage, pneumonia, or microscopic metastasis. Thus, we ensured a net pre-metastatic change with no secondary effects caused by tissue damage. The entire scheme is shown in Fig. 1a. We conducted a laser microdissection to collect fibrinogen-positive and fibrinogen-negative small vessels from non-stained samples. The fibrinogen staining patterns of these vessels were confirmed in serial sections. (Fig. S1a). Fibrinogen-positive areas were observed to have increased in cancer patients, compared to noncancer patients (Fig. S1b). About $2 \times 10^4$ vessels in each group (fibrinogen-positive and -negative in noncancer and cancer patients) were

collected to isolate total RNA. The samples were then subjected to microarray analysis. It was observed that gene expressions of *serum amyloid A* (*SAA*), *S100A8*, and *HMGB1*, reported as mouse pre-metastatic genes[10,11,23], were upregulated in the pulmonary vessels with fibrinogen depositions (Fig. 1b). Of interest, *SAA* genes had 5-fold higher expressions in fibrinogen⁺-vessels in cancer patients as compared to noncancer patients (Supplementary Table 1).

### Generation of humanized SAAs mouse

The human *SAA3* gene encodes only a shorter peptide, and it is not considered to function as a mouse SAA3 protein. Human SAA1 is regarded as a mouse SAA3 counterpart because they share structural and functional similarities[24]. Additionally, it was demonstrated through our array of data, that human *SAA* cluster genes were activated in fibrinogen-deposited vessels in cancer patients. In this study, we replaced the entire region of the mouse *Saa* cluster genes with human genes (Fig. 1a). First, we removed the entire mouse *Saa* cluster genes (mSAAs), constituting *SAA1-4* and *SAAL1* genes in chromosome 7, to obtain mSAAs-KO mice (Fig. 1a, lower, and Fig. S1c). The 70-kb removal from the mouse genome had no apparent effects on their viability, maturation, appearance, or fertility. We also generated transgenic (Tg) mice by inserting 200 kb of human *SAA* cluster genes (*hSAAs*) in them and obtained three lines with one or three copies of the *hSAA* genes (Fig. S1d, e). We then crossed the hSAAs-Tg and mSAAs-KO mice to complete the replacement (Fig. 1a).

To set up the pre-metastatic organ tissues, two conditions, tumor-conditioned media (TCM)-injection and tumor cell implantation techniques, were used. For the tumor cell implantation, we used Lewis Lung Carcinoma (LLC) or E0771 tumor cells, which form primary tumor nodules but do not spontaneously metastasize unless the primary tumor is resected. We use the term "tumor-bearing mouse lungs" to specify "lungs from primary tumor-bearing mice with no metastasis". The "tumor-bearing mouse lungs" also indicate that the lungs are in the pre-metastatic phase. It was demonstrated through RT-PCR analysis that all SAAs were expressed in the livers of wild-type mice but not in SAAs-KO mice (Fig. S1f). Induction of SAA3 and SAAL1 protein expressions were also detected in tumor-bearing mouse lungs (Fig. S1g). To survey the immune system in wild-type, mSAAs-/- and hSAAs/mSAAs-/- mice, we analyzed T cells, B cells, NK (natural killer) cells, dendritic cells, and CD11b⁺ myeloid cells in the peripheral blood taken from these mice, by using flow cytometry. To check the immune responses driven by tumor-derived instigations, these mice were tested with or without tumor-conditioned medium applications. There were no apparent differences in the immune cell populations among these mice (Fig. S1h). Then, we examined the relationships between SAAs and fibrinogen in pre-metastatic lungs using wild-type and mSAAs-KO mice. Immunofluorescence data showed that intense SAA3 and moderate SAA1/2 signals were colocalized with fibrinogen in the lungs of LLC-bearing wild-type mice (Fig. 1c). The SAA3 signal was also observed in the endothelial cells in TCM-stimulated wild-type mice (Fig. 1d). In LLC-bearing mice, the areas of SAA3 and fibrinogen deposition were markedly increased in the wild-type mice. In contrast, the fibrinogen area was slightly increased in SAAs-/- mice (Fig. 1e). An increase in CD45⁺ leukocytes was also observed in LLC-bearing wild-type mice, but not in SAAs-/- mice (Fig. S1i). In studies of 3LL, which is a spontaneous metastatic cell line of LLC, lung metastasis was more severe in wild-type mice, three weeks after implantation, than in SAAs-/- mice (Fig. S1j).

### Citrullination of fibrinogen with SAAs

Next, we targeted the protein−protein interactions between SAA3 and fibrinogen in the tumor-bearing mouse lungs. The fibrinogen molecule is composed of three different polypeptide chains (Aα, Bβ, and γ) to form an (AαBβγ)₂ hexamer[25]. We first detected a faint ~90 kDa band of fibrinogen in lung samples obtained from tumor-bearing and TCM-

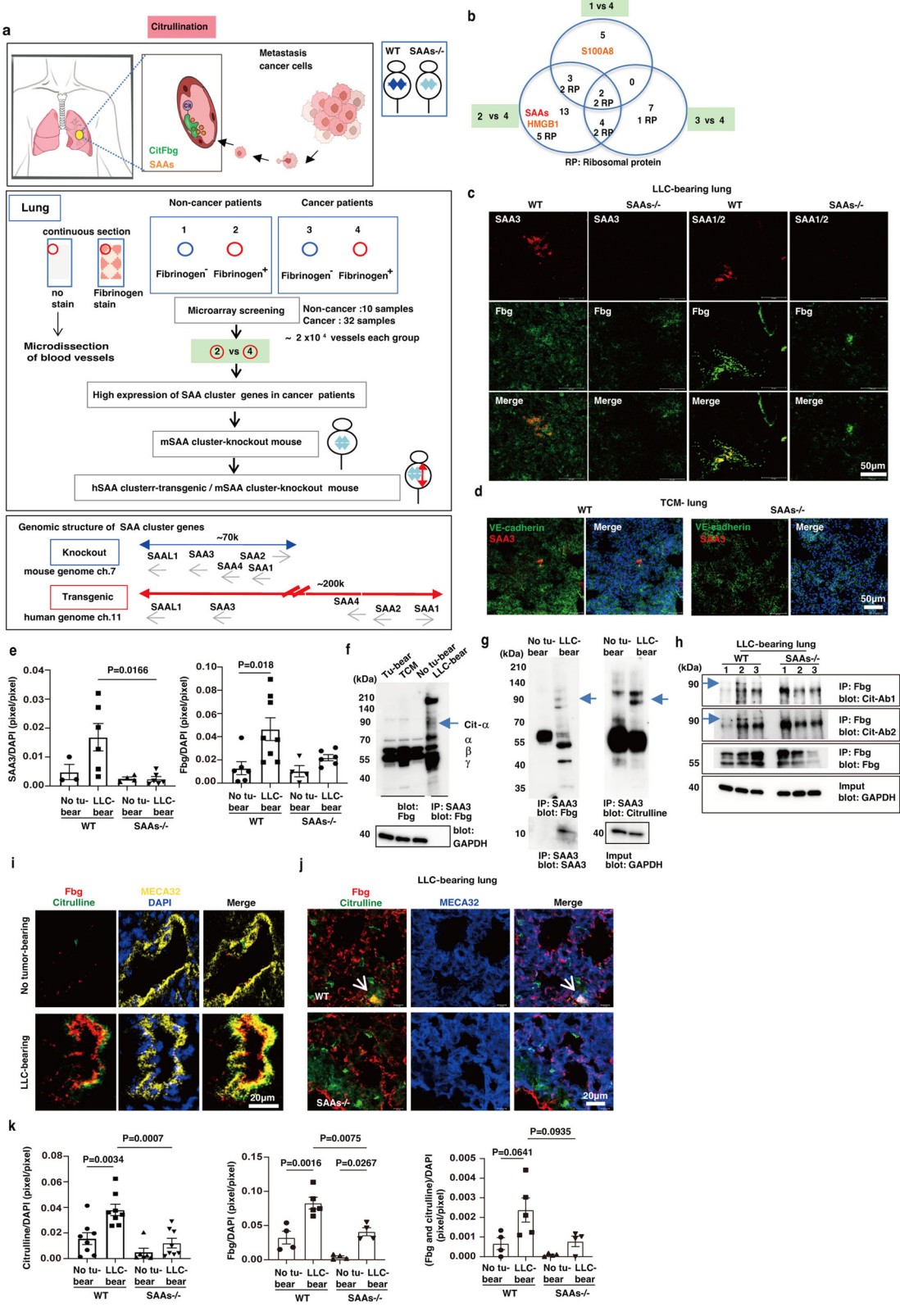

stimulated mice, whereas there was no such counterpart signal in no tumor-bearing mouse lung samples (Fig. 1f). The 90 kDa band, as well as three fibrinogen subunits, were clearly detected in the immuno-precipitation product of lung lysate from tumor-bearing condition by anti-SAA3 antibody (Fig. 1f, right lane). This data implied that there was a SAA3-fibrinogen interaction with the 90 kDa component. We again confirmed the upshifted fibrinogen band in the SAA3-binding

precipitate using lung lysate derived from tumor-bearing mice. The lung lysate obtained from the no tumor-bearing mice contained undetectable levels of SAA3, so the anti-SAA3 pull-down produced no SAA3 band (Fig. 1g, left). This SAA3-negative lane did not show bands unique to fibrinogen, suggesting that the upshifted fibrinogen band could not be nonspecifically detected. The reason for this shift was assumed to be a post-translational modification in the Aα chain

**Fig. 1 | SAAs-dependent citrullination of Fbg. a** Summary model depicting the occurrence of lung metastasis at the citrullination site in cancer patients (upper). Replacement of *Saa* cluster genes in the mouse genome by a human counterpart (middle). Screening of genes related to the pre-metastatic phase using Fbg-deposited pulmonary vessels from noncancer and cancer patients. Generation of murine SAA gene cluster deficient and human SAA gene cluster transgene mice. Genomic structure of mouse and human SAA cluster genes (bottom). **b** Summary of microarray data. Venn diagram of genes with an over 4-fold increase in mRNA expressions comparing No. 1 vs. No. 4, No. 2 vs. No. 4, and No. 3 vs. No. 4 depicted in Fig. 1a. Gene names and accession numbers for No. 2 vs. No. 4 are shown in supplementary Table 1. **c** SAA3 and SAA1/2 proteins were colocalized with Fbg in LLC-bearing mouse lungs. Scale bar, 50 μm. **d** Representative immunohistochemistry (IHC) photo of SAA3 expression in endothelial cells (EC) of tumor-conditioned media (TCM)-stimulated mouse lungs. Scale bar, 50 μm. **e** SAA3 induction in LLC-bearing-wild-type but not -SAAs−/− mice (left) ($n = 3$ WT-no tumor; $n = 6$ WT-tumor; $n = 4$ SAAs−/−-no tumor; $n = 6$ SAAs−/−-tumor), and Fbg induction (right) ($n = 6$ WT-no tumor; $n = 8$ WT-tumor; $n = 4$ SAAs−/−-no tumor; $n = 6$ SAAs−/−-tumor). One-way

ANOVA with Bonferroni correction. Mean ± SEM. **f** Appearance of a shifted band of Fbg in the total lysate and SAA3-IP samples prepared from primary tumor-stimulating lungs. Arrow shows a citrullinated α chain of Fbg. Two independent experiments. **g** Fbg bound to SAA3 in LLC-bearing lungs was assumed to be citrullinated (arrow). Immunoprecipitated SAA3 (left bottom) was also presented in the western blot. Two independent experiments. **h** The citrullinated Fbg band in tumor-bearings in a SAAs-dependent manner (arrow). Two anti-citrullination antibodies were used. Three independent experiments. **i, j** Immunohistochemical colocalization signal between citrulline and Fbg on MECA32+-endothelial cells in LLC-bearing mouse lungs (**i**) and the signal was prominent in wild-type compared to SAAs−/− mice (arrowhead). Scale bar, 20 μm. **k** Immunohistochemical quantification of citrulline (left) ($n = 8$ WT-no tumor; $n = 8$ WT-LLC tumor; $n = 6$ SAAs−/−-no tumor; $n = 8$ SAAs−/−-LLC tumor), Fbg (middle), and citrullinated Fbg (right) ($n = 4$ WT-no tumor; $n = 5$ WT-LLC tumor; $n = 4$ SAAs−/−-no tumor; $n = 4$ SAAs−/−-LLC tumor) in no tumor-bearing- and LLC-bearing-wild-type and SAAs-/- mouse lungs. One-way ANOVA with Bonferroni correction. Mean ± SEM.

(Fig. 1f). So far, 11 types of PTM on fibrinogen have been reported. These modifications affect clot formation, clot characteristics, and susceptibility fibrinolysis[26]. Among these, the shift of the Aα chain band suggested the presence of citrullination modifications[27], which was confirmed by anti-citrulline blotting (Fig. 1g right). Additionally, citrullinated fibrinogen was clearly detected in LLC-bearing mouse lungs, whereas it was modest in tumor-bearing SAAs-/- mouse lungs (Fig. 1h, hereinafter, fibrinogen and citrullinated fibrinogen are described as "Fbg" and "CitFbg," respectively). The citrullinated residues were detected in TCM-stimulated and tumor-bearing mouse lungs using immunohistochemical analysis (Fig. S1k). Later, we examined depositions of CitFbg in pulmonary vessels because metastatic tumor cells tend to be deposited on the vascular bed generated in the pre-metastatic phase[28]. The colocalized signals of citrullination and Fbg in small vessels were surrounded by endothelial cells in tumor-bearing mouse lungs (Fig. 1i and Fig. S1l, arrowhead). CitFbg serves as a metastatic niche and is located in distinct areas of tumor-bearing wild-type mouse. On the contrary, it was quite rare in tumor-bearing SAAs-/- mice (Fig. 1j, k, right graph).

Next, we wanted to identify the cells responsible for the citrullination of Fbg in primary tumor-stimulating lung microenvironments. In the tumor-bearing mouse lungs, wild-type mice have more CD11b+ myeloid cells than SAAs-/- mice (Fig. S2a). We also examined PAD2 and PAD4 in these tissues because Fbg citrullination may be attained by these enzymes. The immunohistochemical analyses revealed that PAD4 expression in endothelial cells and CD11b+ cells were clearly increased in tumor-bearing wild-type mice in a SAAs-dependent manner (Fig. 2a). PAD2 expression data showed similar increases, although they were not statistically significant (Fig. S2b). To investigate the enzymatic abilities of PADs in those cells, we isolated CD144+, CD11b+, and F4/80+ cells from tumor-bearing wild-type and SAAs-/- mouse lungs to apply them onto Fbg-coated wells for our in vitro assay (Fig. 2b). This assay allowed us to test if these cells possessed citrullination ability without assistance from any other cells (Fig. S2c). Up to 100 μg of Fbg was coated in the wells, and citrullination was quantified by western blot (Fig. 2c and Fig. S2d) to determine the extent of the citrullination reaction in each cell. The CD144+ endothelial cells were seen to have more vigorous citrullination activity than CD11b+ and F4/80+ cells. It should be noted that cells from SAAs-/- mice had a trace amount of activity in this assay (Fig. 2c, d and Fig. S2e). To confirm PAD2- and PAD4- dependent citrullination on Fbg mediated by endothelial cells and myeloid cells derived from tumor-bearing mice, we carried out siRNA transfection into these cells. As shown in Fig. 2e, the knockdown of both PAD2 and PAD4 suppressed the citrullination of Fbg, suggesting that both these enzymes participate in Fbg citrullination in the case of lung ECs. It was also observed that PAD4 knockdown in lung CD11b+ myeloid cells reduced citrullination of Fbg (Fig. 2e), as

expected from the Fig. 2a data. We further searched for factors that would stimulate PAD-protein expression using our lung organ culture system. We tested candidate molecules reported to be pre-metastatic inducer, such as mSAA3, hSAA1, S100A8, CCL2, SDF1, and VEGF, to find out that mSAA3 significantly induced PAD2 and PAD4 in mouse lungs. Recombinant hSAA1 was also effective in increasing PAD2 expression (Fig. S2f).

## Citrullination triggers metastasis

We have previously measured the distance between metastatic tumor cells and hyperpermeable vessels having Fbg deposition in the lungs of tumor-bearing mice. This hyperpermeability region was visualized by fluorescent-labeled microbeads, and the data demonstrated that the tumor cells in the lung parenchyma stayed within 200 μm from the leaky vessel, 24 h after the tail vein injection[28]. We modified this assay to clarify the relationship between Fbg citrullination and tumor cell extravasation. In this assay system, since tumor cells injected via the tail vein mimic circulating tumor cells, the status before and after the injection into tumor-bearing mice are defined as pre-metastatic and post-metastatic phases, respectively (Fig. 2f, upper).

After the injection, we measured the distance between CitFbg/SAAs and fluorescent-labeled tumor cells in the post-metastatic lungs (Fig. 2f, lower). First, the wild-type mice harbored more lung metastatic cells than SAAs-/- mice (Fig. 2g). It was assumed that SAAs provide a conductive environment to the circulating tumor cells. We performed immunohistochemical analyses of LLC-bearing post-metastatic lungs using SAA1/2, SAA3, Fbg, and citrullinated peptide antibodies to obtain detailed information (Fig. 2h and Fig. S2g). Notably, the tumor cells were observed to be close to SAA1/2, SAA3, citrullination, and Fbg (Fig. 2h). In contrast, there were few signals of citrullination and Fbg near the tumor cells in the case of LLC-bearing SAAs-/- mice (Fig. S2g). The E0771-bearing mice gave similar results to the LLC-bearing mice (Fig. S2h). Finally, as shown in Fig. 2i, SAA3, citrullination, Fbg, and metastatic tumor cells were situated in almost the same location. This result indicates that the coexistence of SAA3 and CitFbg might be an estimated marker of the metastatic site. Next, we calculated the signals of each molecule normalized by DAPI in square regions near the tumor cells (Fig. S2i). It was shown in the data that SAA3, SAA1/2, and CitFbg were present not far from the metastatic tumor cells (Fig. 2j and Fig. S2j). There seemed to be no correlation between CD11b+ myeloid cells and CitFbg hotspots (Fig. S2k). As CD11b+ myeloid cells can freely move in the lung tissue, they do not have to be positioned near SAA-CitFbg, even if they have a functional relationship with SAA-CitFbg. Of interest, around 3% of Fbg was found to be citrullinated in the hyperpermeable Fbg-rich area near the metastatic tumor cells (Fig. 2j, right). This rate was much higher than in the other tumor cell-free areas. In addition, the CitFbg/Fbg ratio-distance pattern of wild-type mouse was

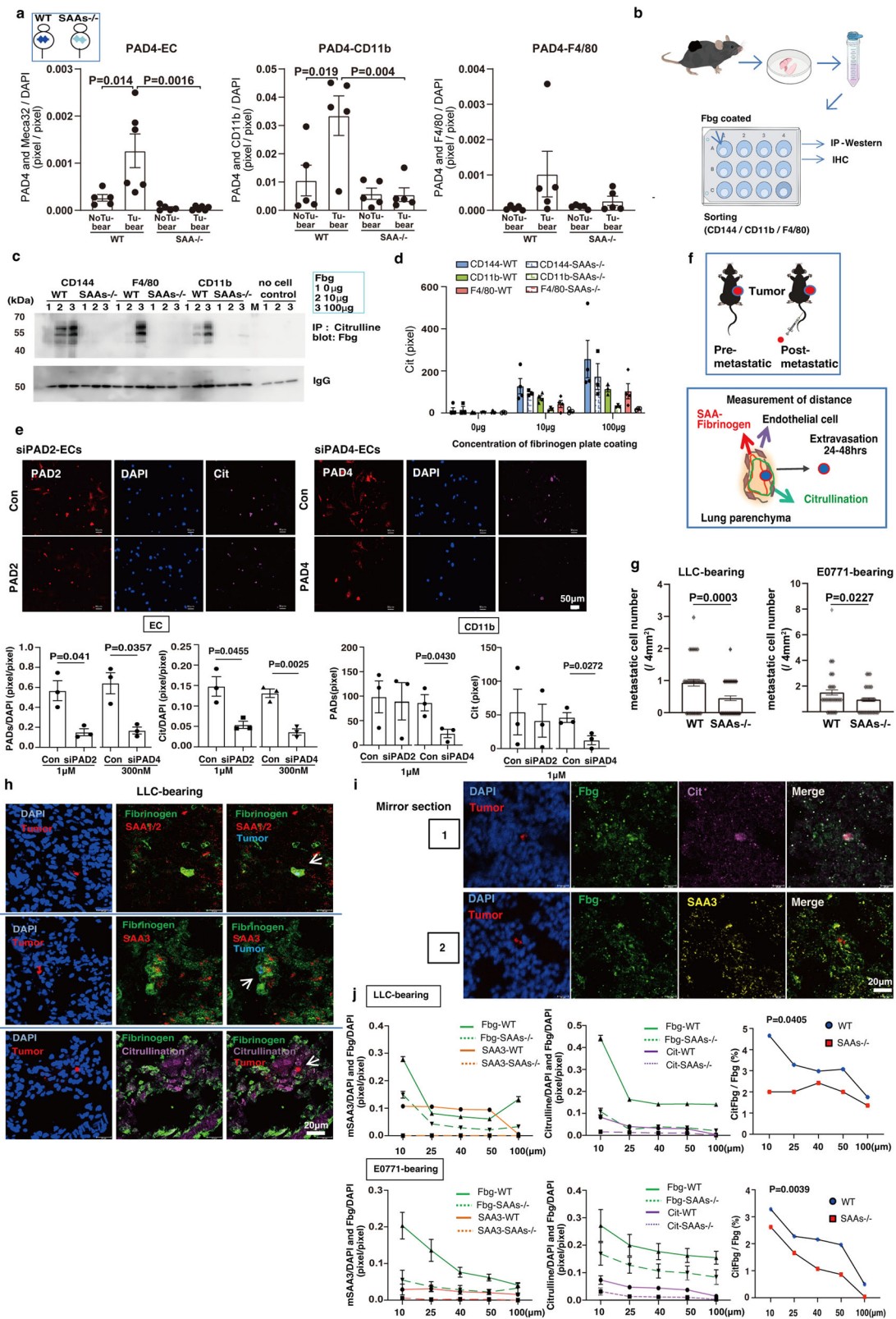

different from that of SAAs-/- mouse (Fig. 2j, right). With these results, we further investigated the cooperative function of SAAs-CitFbg in metastasis.

We injected hCitFbg into normal wild-type and SAAs-/- mice to examine whether SAAs-CitFbg was peculiar to the pre-metastatic niche formation. In a mouse study, it had been demonstrated that 5 mg/kg of human Fbg (hFbg) injection caused blood-brain barrier leakage,

leading to neuroinflammatory disease[29]. We tested several concentrations of hCitFbg protein injections into mice based on these data and found out that 2 μg/kg (50 ng/animal) was the optimum concentration, since it was close to the concentration in human serum[30]. In this assay, after an intravenous injection of either control PBS, hFbg, or hCitFbg, labeled tumor cells were injected into the wild-type mice and SAA-/- mice (Fig. 3a). As expected, pre-injected hCitFbg strongly

**Fig. 2 | Metastatic tumor cells located close to SAAs-CitFbg. a** Quantifications of PAD4 in endothelial cells (EC) (PAD4-EC) (*n* = 5 WT-no tumor; *n* = 6 WT-LLC tumor; *n* = 5 SAAs-/--no tumor; *n* = 6 SAAs-/--LLC tumor), CD11b⁺ cells (PAD4-CD11b) (*n* = 5 WT-no tumor; *n* = 5 WT-LLC tumor; *n* = 5 SAAs-/--no tumor; n = 5 SAAs-/--LLC tumor) and F4/80⁺ cells (PAD4-F4/80) (*n* = 5 WT-no tumor; *n* = 5 WT-LLC tumor; *n* = 5 SAAs-/--no tumor; *n* = 5 SAAs-/--LLC tumor) in no tumor-bearing and LLC-bearing lung tissues in IHC analysis. One-way ANOVA with Bonferroni correction. Mean ± SEM. **b** Cell-mediated citrullination assay scheme. The same number of CD144⁺ EC, CD11b⁺, and F4/80⁺ cells derived from tumor-bearing lungs were applied on various concentrations of Fbg-coated plates. Fbg citrullination was evaluated by IP-western blot (Fig. 2c). **c** Citrullination ability in purified EC compared with CD11b⁺ cells and F4/80⁺ cells. Three independent experiments. **d** Measurement of citrullination area of Fbg-coated plate by purified cells from tumor-bearing-wild-type and SAAs−/− mouse lungs. Mean ± SEM. Three independent experiments. **e** Citrullination after knockdown of PAD2 and PAD4 in ECs and CD11b⁺ cells derived from tumor-bearing mouse lungs. Immunohistochemical image of ECs (upper) and quantification of PAD and citrullination signals normalized by DAPI (ECs). To calculate the PADs signals of CD11b⁺ cells, DAPI normalization was omitted because extracellular PADs from these cells should be considered (lower). Mean ± SEM (Student's two-sided *t*-test). Scale bar, 50 μm. Three independent experiments. **f** Pre-metastatic and post-

metastatic models (upper). Mouse-holding tumors that did not spontaneously metastasize were defined as pre-metastatic. A pre-metastatic mouse that received an injection of metastatic tumor cells in the blood vessel was defined as post-metastatic. The short distance between post-metastatic cells and SAAs-CitFbg implies a presumptive pre-metastatic niche (lower). **g** Number of metastatic cells in lungs after the tumor cell injection to LLC-bearing and E0771-bearing mice. For LLC-bearing mouse and E0771-bearing mouse analyses, 60 and 56 sections of WT and SAAs−/− lungs were examined, respectively. In the analysis, two to three sections per lobe were chosen, and all lobes (from five WT and five SAAs−/− mice) were examined. Mean ± SEM (Student's two-sided *t*-test). **h** Recruitment of tumor cells around SAA1/2-Fbg, SAA3-Fbg, and CitFbg in LLC-bearing mice. Tumor cells can be identified by the blue pseudo-color in merged SAA signals. Scale bars, 20 μm. Six independent experiments. **i** Mirror section to detect tumor cells near the SAA3-CitFbg signals in the lung. IHC analysis for two consecutive frozen sections, No. 1, and No. 2, with the cut surfaces facing each other upwards for staining with different antibodies. Scale bars, 20 μm. Two independent experiments. **j** Kinetic distance between tumor cells and proteins. Data analysis based on a staining set of SAA3/Fbg (left), citrulline/Fbg (middle), and CitFbg/Fbg (right) in tumor-bearing wild-type and SAAs−/− mice (areas were chosen from *n* = 5 wild-type mice and from *n* = 5 SAAs−/− mice, respectively). Mean ± SEM. Two-way ANOVA.

---

attracted metastatic tumor cells in wild-type as well as SAAs-/- mice (Fig. 3b). Additionally, SAAs might appear to have supported CitFbg-mediating metastasis, because wild-type mice had a higher degree of metastasis than SAAs-/- mice (Fig. 3b). Three weeks later, the group that had received a pre-injection of CitFbg showed more severe metastasis compared to PBS control group (Fig. 3c).

We established two hybridoma cell lines producing monoclonal antibodies (clones A-s1 and F) against hCitFbg in this study. We validated these antibodies by trying them against Fbg-overexpressing Chinese hamster ovary (CHO) cells treated with the catalytic enzyme PAD4 (Fig. S3a). The specificities of clone A-s1 and clone F antibodies were confirmed to be the alpha and beta of hCitFbg, respectively (Fig. S3b, c). The clone A-s1 antibody was detected on tumor cells in blood vessels, where hCitFbg depositions were formed on ECs in the hCitFbg-injected wild-type mice (Fig. 3d, lower and Fig. S3d). CD11b⁺ cells were diffusely observed around the hCitFbg depositions (Fig. 3d, upper). CitFbg depositions around metastatic tumor cells resulted in intense CitFbg signals. These signals were even enhanced in the presence of SAA, suggesting that SAAs may promote the citrullination of hFbg (Fig. 3e and Fig. S3e). This theory is supported by the data in Fig. 3f, which shows the distance between the tumor cells and hCitFbg.

We successfully proved that innate immune cells, including CD11b⁺ and F4/80⁺ cells, contributed to the citrullination of Fbg. Conversely, the pre-metastatic microenvironment contained adaptive immune cells, such as T and B cells[5,31,32]. Therefore, we next decided to test whether T cells and B cells were involved in the pre-metastatic citrullination of Fbg using NOD/Scid mice (Fig. 3g). NOD/Scid mice also had hCitFbg-stimulated mobilization of tumor cells into the lungs (Fig. 3h), indicating that there was no difference between Scid and C57BL/6 mice. In the lungs of the hCitFbg-injected mice, SAA3 induction was not observed unless metastatic tumor cells were nearby (Fig. 3i). Finally, data showing the distance between tumor cells and CitFbg, Fbg, or SAA3 revealed that CitFbg-SAA3 accumulated in metastatic tumor microenvironments. (Fig. 3j).

### Detection of CitFbg in cancer patients

It is necessary to explore a new molecular tool to determine potential metastatic sites accurately. With this end in mind, we immunohistochemically evaluated our antibodies against CitFbg in tissues from patients. It is a known fact that in rheumatoid arthritis (RA) patients, accumulate citrullinated proteins, including Fbg, accumulate in their rheumatoid joints[33]. After we compared the staining patterns of the slides prepared from RA and non-RA (diagnosed as not RA) patients, we determined that our clone A-s1 antibody (alpha chain-specific,

referred to hCitFbg-α-Ab) detected CitFbg without cross-reaction for massive Fbg/fibrin, while the clone F antibody did not (Fig. S4a). We then applied this hCitFbg-α-Ab staining to no-metastatic lungs derived from noncancer and cancer patients (Fig. 4a, b). These samples were not the same samples used for screening in Fig. 1 to avoid sample bias. In the immunostaining, it was observed that the CitFbg deposition in the pulmonary vascular bed and the bronchovesicular region was far less than the Fbg deposition (Fig. 4c and Fig. S4b). The CitFbg deposits in the blood vessels were more prevalent in cancer patients than in noncancer patients (Fig. 4d–f, i). Moreover, the colocalization signals of CitFbg and SAA1/2 were enhanced in cancer patients (Fig. 4g, h). We further investigated the relationship between metastasis and CitFbg deposition in metastatic lungs (Fig. 4j). As shown in Fig. 4k, the no-metastatic region in the metastatic lobe (blue square) showed more CitFbg signals than the no-metastatic lobe (orange circle). The tumor cell-free area along the metastatic site recorded the highest signals of CitFbg (Fig. 4l–n, q) and SAA-CitFbg (Fig. 4o, p).

We studied our *SAAs*-humanized mice thoroughly (Figs. 1a, 5a) to examine molecules that could act as potential tumor cell attractants in cancer patients. It was observed that lungs derived from TCM-stimulated wild-type (mSAAs+/+) and hSAAs-Tg /mSAAs-/- mice contained almost the same number of CD11b⁺ cells, suggesting that these mice elicit similar responses of myeloid cells in the pre-metastatic stage (Fig. S5a). These mice did not express SAA proteins in a steady state, however, after CitFbg was injected, they were seen to express these proteins (Fig. 5b). Immunostainings of hCitFbg-injected mouse lungs were suggestive of the fact that hSAA1 and hSAA4 were in close proximity to ECs and CD11b⁺ cells, and mSAAs-KO mouse data indicated that these hSAA1 and hSAA4 signals were distinctive (Fig. S5b). We next examined whether the hSAAs-hCitFbg protein complex was responsible for the niche for metastatic tumor cells. The numbers of human metastatic tumor cells were observed to be remarkably elevated in hCitFbg-treated hSAAs-Tg/mSAAs-KO mice compared to mSAAs-KO mice (Fig. 5c). In contrast to hCitFbg, only a small fraction of metastatic tumor cells were detected in Fbg-treated hSAAs-Tg/mSAAs-KO mice (Fig. 5c). As the injection of hCitFbg was accompanied by the induction of hSAA1 and hSAA4 (Fig. S5c), the complex formation of hSAAs-hCitFbg might be directly interacting with metastatic tumor cells. The hCitFbg-treated hSAAs-Tg/mSAAs-KO mouse lungs had strong signals of the hSAAs-hCitFbg complex near the labeled tumor cells (Fig. 5d and S5d). In contrast, the signals of hCitFbg were faint in the case of the hCitFbg-treated mSAAs-KO mice (Fig. 5d and S5d). Distances between the hCitFbg of the complex and tumor cells were shorter in hCitFbg-treated hSAAs-Tg/mSAAs-KO mice (Fig. 5e and S5e).

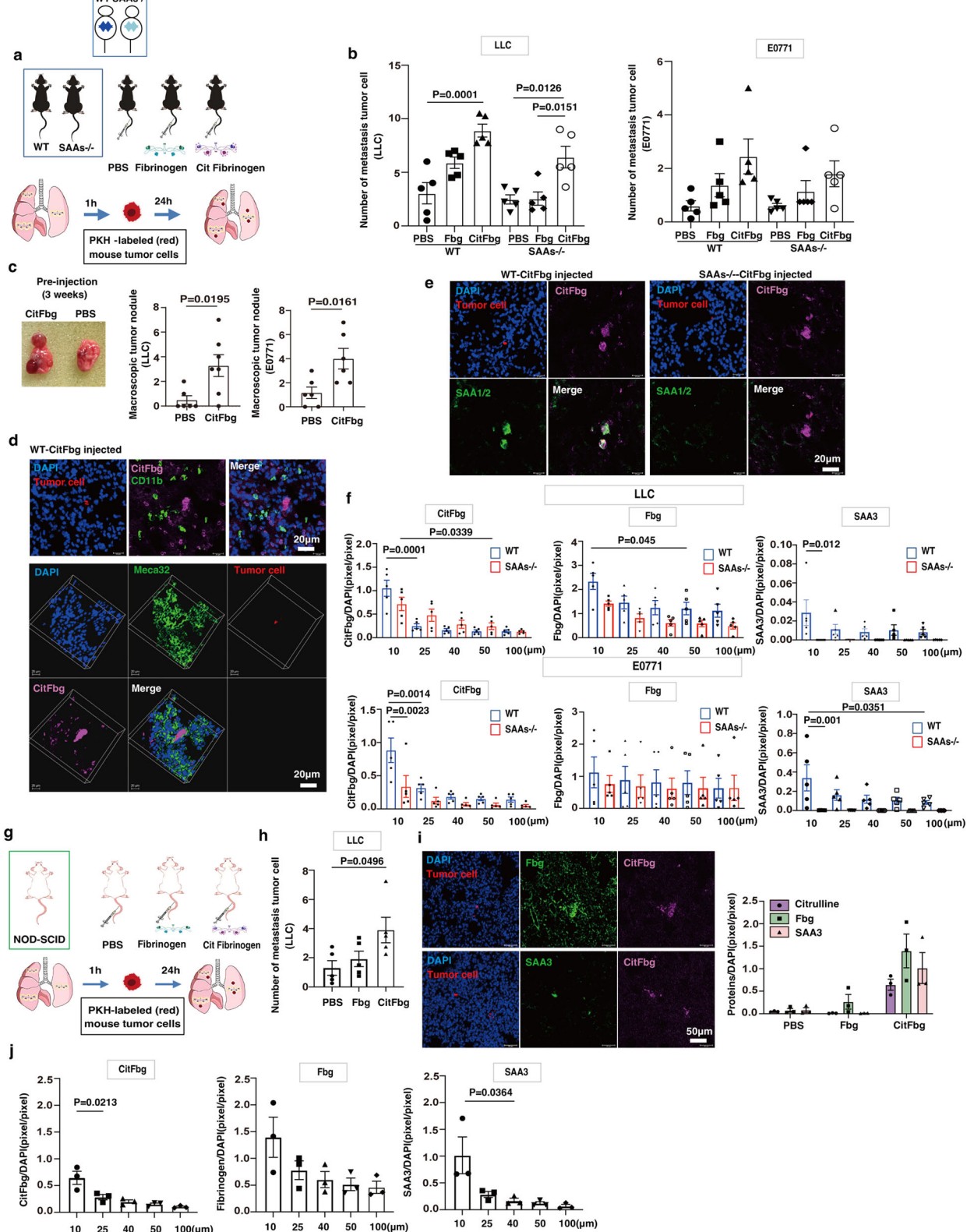

However, there was no such correlation observed in the hCitFbg-treated mSAAs-KO mice (Fig. 5e and S5e). Thus, the hSAA1-hCitFbg complex was an indicator of the metastatic risk area. Our in vivo data demonstrated that hSAA1-CitFbg and hSAA4-CitFbg complexes were located in the 20 μm range from the metastatic human tumor cells (Fig. 5f and S5f). In mice, mSAA3 functioned as a binding partner molecule of CitFbg in the pre-metastatic milieu instead of hSAA1 and

hSAA4 (Fig. 5f). Finally, we examined whether the entrapment of circulating tumor cells by hSAAs-hCitFbg effectively led to parenchymal metastasis. We observed lung metastasis 3 weeks after the injection of human cancer cells, MDAMB231 or MCF7, into mice with Rag1-/- background. The results indicated that hSAAs/mSAAs-/-Rag1-/- mice had more metastatic tumor cells in the lungs than mSAAs-/-Rag1 -/- mice (Fig. 5g and S5g).

**Fig. 3 | Human citrullinated Fbg (hCitFbg) attracted tumor cells in the presence of SAAs. a** Pre-injection of PBS, hFbg, and hCitFbg in wild-type and SAAs-/- mice on the C57BL/6 background. These mice received an injection of labeled tumor cells. The lung tissues were then analyzed. An anti-hCitFbg (clone A-s1, see Fig. S3) was used to detect injected hCitFbg. **b** Number of metastatic LLC and E0771 cells 24 h after the injection in wild-type and SAAs-/- mice pretreated with PBS, hFbg, or hCitFbg. ($n = 5$ WT; $n = 5$ SAAs-/- mice) One-way ANOVA with Bonferroni correction. Mean ± SEM. **c** Metastatic nodules 3 weeks after tumor cell injection in wild-type mice pretreated with PBS or hCitFbg (left). Quantification of nodule number ($n = 5$, right). Mean ± SEM (Student's two-sided *t*-test). **d** Tumor cells and CD11b[+] cells showing hCitFbg in wild-type mice (upper). Migration of tumor cells to the hCitFbg deposition site on MECA32[+] EC in pulmonary vessels is shown in the stacked 3D image (lower). Scale bars, 20 μm. Three independent experiments. **e** Metastatic tumor cells near CitFbg in hCitFbg-treated wild-type and SAAs-/- mice. Scale bars,

20 μm. Five independent experiments. **f** Distances among the tumor cells and hCitFbg, Fbg, and SAA3 in hCitFbg-treated wild-type and SAAs-/- mice. ($n = 5$ WT; $n = 5$ SAAs-/- mice) One-way ANOVA with Bonferroni correction. Mean ± SEM. **g** Scheme of hCitFbg-induced metastasis assay similar to Fig. 3a in NOD/Scid mouse background. **h** The number of metastatic LLC cells in the lung tissues derived from NOD/Scid mice pretreated with PBS, hFbg, or hCitFbg. ($n = 5$). One-way ANOVA with Bonferroni correction. Mean ± SEM. **i** Tumor cells on hCitFbg in diffuse Fbg (upper panel) and tumor cells close to hCitFbg accompanied by SAA3 (lower panel). Scale bars, 50 μm. Quantitative analysis of citrulline, Fbg, and SAA3 around tumor cells (within 20 μm) in NOD/Scid mice pretreated with PBS, hFbg, or hCitFbg (right). ($n = 3$ NOD/Scid). One-way ANOVA with Bonferroni correction. Mean ± SEM. **j** Distances among metastatic LLC cells, hCitFbg, Fbg, and SAA3 were shown. ($n = 3$ NOD/Scid). One-way ANOVA with Bonferroni correction. Mean ± SEM.

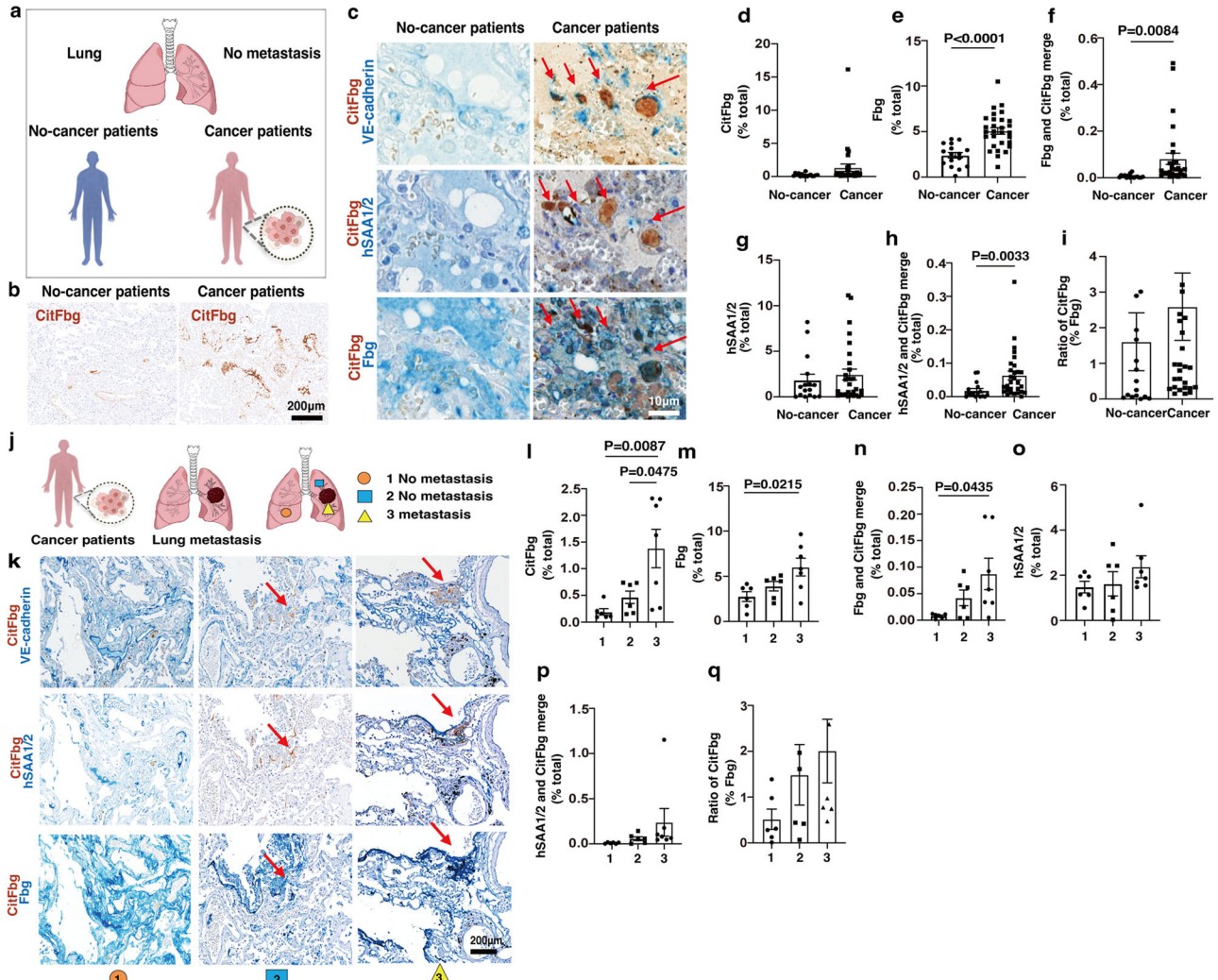

**Fig. 4 | A retrospective study clarified the deposition of SAA1/2-CitFbg in the lungs of cancer patients. a** IHC analysis of CitFbg in the alveolar region of non-cancer and cancer patients with a specific antibody for hCitFbg that was validated to have no cross-reaction for fibrin. Lung tissue samples with no microscopical metastases were used. **b** The DAB signals in capillaries of cancer patients were shown. Scale bars, 200 μm. Four independent experiments. **c** Representative images with double staining of CitFbg/VE-cadherin and CitFbg/SAA1/2 and CitFbg/Fbg. Arrows indicate the same vessels. Scale bars, 10 μm. **d–h** Quantification of the area of CitFbg (**d**), Fbg (**e**), CitFbg/Fbg (**f**), SAA1/2 (**g**), and CitFbg/SAA1/2 (**h**) signals in small pulmonary vessels ($n = 16$ no cancer, $n = 29$ cancer). Data were mean ± SEM

(Student's two-sided *t*-test). **i** Ratio of the area of CitFbg in Fbg ($n = 16$ no cancer; $n = 29$ cancer). Data were mean ± SEM (Student's two-sided *t*-test). **j** IHC analysis using various regions with no evidence of tumor cells derived from no-metastatic lobes, no-metastatic, or metastatic regions in lobes with metastasis in the same cancer patients. **k** Representative images of CitFbg/VE-cadherin, CitFbg/SAA1/2, and CitFbg/Fbg. Arrows indicate the same position. Scale bars, 200 μm. **l–p** Quantification of the area of CitFbg (**l**), Fbg (**m**), CitFbg/Fbg (**n**), SAA1/2 (**o**), and CitFbg/SAA1/2 (**p**). **q** Ratio of the area of CitFbg in Fbg ($n = 6$ no metastasis; $n = 7$ metastasis). One-way ANOVA with Bonferroni correction. Data were mean ± SEM.

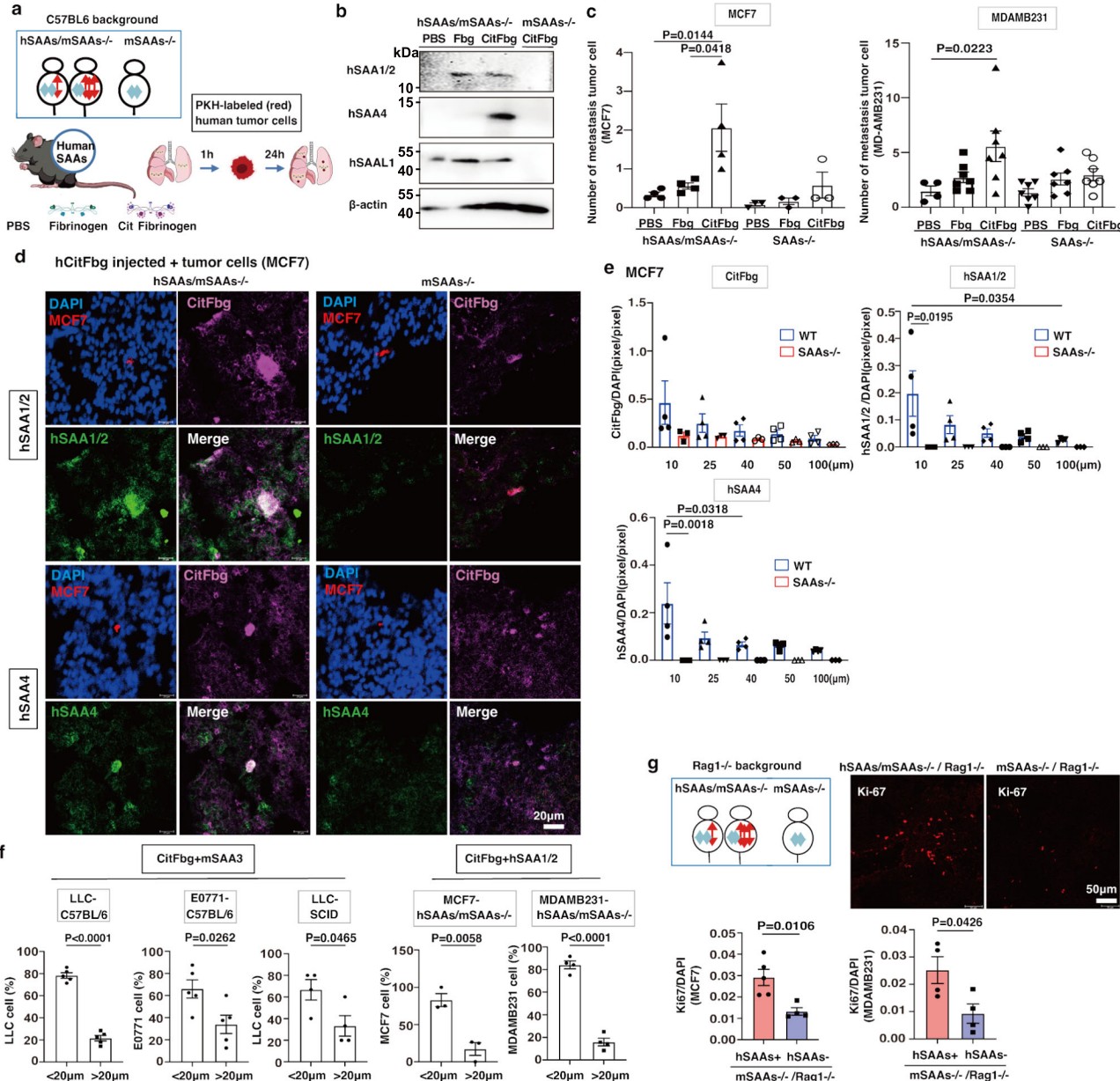

**Fig. 5 | Prospective study for proving that humanized molecular complex in pre-metastatic lungs trapped human cancer cells. a** Scheme of hSAAs-hCitFbg complex-induced metastasis. First, PBS, hFbg, or hCitFbg was injected into humanized hSAAs-Tg/mSAAs-KO mice. Then, labeled human cancer cells were injected. **b** Detection of SAAs proteins in lung tissues after injecting hCitFbg into hSAAs/mSAAs-/- mice by western blot analysis. The mSAAs-/-mice were used as a negative control. Two independent experiments. **c** Numbers of metastatic MCF7 (*n* = 4 hSAAs/mSAAs-/-; *n* = 3 SAAs-/-) and MDAMB231 cells (*n* = 4 hSAAs/mSAAs-/--PBS; *n* = 7 hSAAs/mSAAs-/--hFbg and CitFbg; *n* = 7 SAAs-/-) in PBS, hFbg-, and hCitFbg-treated-hSAAs/mSAAs-/- and mSAAs-/-mice. One-way ANOVA with Bonferroni correction. Data were mean ± SEM. **d** Representative IHC photos of metastatic MCF7 cells trapped by the hSAAs·hCitFbg complex. Scale bars, 20 μm. **e** Distances among

hCitFbg, hSAA1/2, hSAA4, and MCF7 cells shown in (**d**) were analyzed (*n* = 4 hSAAs/mSAAs-/-; *n* = 3 mSAAs-/-) One-way ANOVA with Bonferroni correction. Data were mean ± SEM. **f** Summary of the distance among metastatic tumor cells, SAA3-hCitFbg, and hSAA1/2-hCitFbg. (*n* = 5 LLC-C57BL/6; *n* = 5 E0771-C57BL/6; *n* = 4 LLC-Scid; *n* = 3; MCF7- hSAAs/mSAAs-/-; *n* = 4 MDAMB231- hSAAs/mSAAs-/-) Data were mean ± SEM (Student's two-sided *t*-test). **g** Similar strategy to Fig. 5a in hSAAs-Tg/mSAAs-KO in Rag1-/- background (upper left). Representative staining of Ki67 in lung 3 weeks after injection of tumor cells in hSAAs/mSAAs-/-/Rag1-/- and mSAAs-/-/Rag1-/- mice (upper right). Quantification of metastatic surface area in lungs 3 weeks after application of MCF7 (*n* = 5 mSAA-/-hSAAs-Tg/-Rag1-/-; *n* = 4 mSAA-/-hSAAs-/-Rag1-/-) and MDAMB231 (*n* = 4 mSAA-/-hSAAs-Tg/-Rag1-/-; *n* = 4 mSAA-/-hSAAs-/-Rag1-/-) cells (lower). Data were mean ± SEM (Student's two-sided *t*-test).

## Citrullination sets up a metastatic site

We conducted the following experiments to confirm and characterize the complex formation of SAA-Fbg or SAA-CitFbg. Citrullination converts the guanidino group to the ureido group, and this results in the loss of a positive charge, because of which, hCitFbg possesses fewer positive charges than hFbg. This was confirmed by the higher NaCl concentration required for the elution of hCitFbg as compared to hFbg from an anion exchange column (Fig. S6a). Additionally, this data

confirmed that the conversion of hFbg to hCitFbg occurred almost completely. Enzyme-linked immunosorbent assay (ELISA) revealed that hSAA1 binds to hFbg and hCitFbg. Apparent Kd value of hSAA1-Fbg was half for hSAA1-CitFbg as compared to hSAA1-Fbg; at the same time, the amount of hSAA1 binding (Bmax) on hCitFbg was twice as much as that on hFbg (Fig. 6a). Thus, hSAA1 binding to hFbg and hCitFbg is similar in conditions of low hSAA1 concentrations (<30 ng/mL), as shown in our ELISA data (Fig. 6a). The association of SAA with

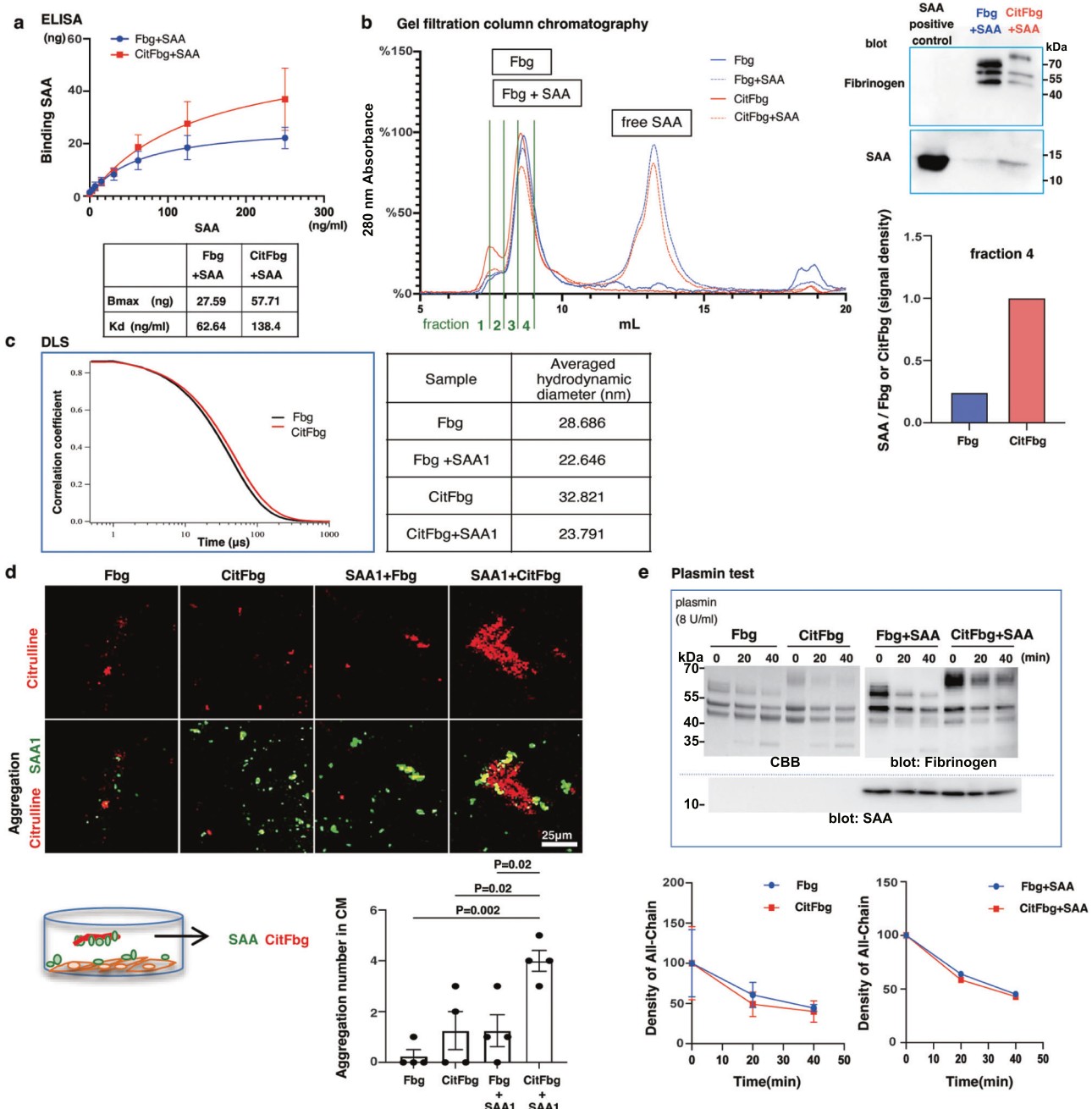

**Fig. 6 | Citrullination changes the structure of the human Fbg-SAAs complex.**
**a** Enzyme-linked immunosorbent assay (ELISA) for the binding between hSAA1 and hFbg or between hSAA1 and hCitFbg. Data points show mean ± SEM of four independent experiments. **b** Gel filtration analysis of hFbg and hCitFbg in the absence and presence of hSAA1 (left). The eluted fraction (fraction 4) corresponding to hSAA1-hFbg and hSAA1-hCitFbg was analyzed using a western blot (right). The intensities of the bands corresponding to hFbg, hCitFbg, and hSAA1 were quantified (lower). Three independent experiments. **c** Dynamic light scattering (DLS) analysis of Fbg and CitFbg in the presence and absence of SAA1 (left). The averaged hydrodynamic diameters were listed (right). **d** Formation of aggregations in CM after application of Fbg, CitFbg, and SAA1 in the presence of human lung microvessel ECs (lower left). Representative images of protein aggregations (upper). Large-size aggregations were counted in comparison with those of naturally secreted SAA1 from ECs (lower right). One-way ANOVA with Bonferroni correction. Data were mean ± SEM. Scale bars, 25 μm. Four independent experiments. **e** Degradation kinetics of Fbg, CitFbg, SAA1-Fbg, and SAA1-CitFbg with plasmin were shown in PAGE-Coomassie blue staining (upper left) and Western blot analysis (upper right, reducing condition). Comparison of degradation rates of Fbg vs. CitFbg (Data points show mean ± SEM of two independent experiments) and SAA1-Fbg vs. SAA1-CitFbg (lower).

Fbg or CitFbg was also supported by the analysis of size exclusion chromatography (Fig. 6b). The elution profiles of hFbg and hCitFbg had significant peaks at 8.7 mL, reflecting their large molecular weights. The broad elution peak for hSAA1 alone was located at 13.5 mL, indicating that it had a smaller molecular weight along with possible oligomerization. The profiles for the samples obtained after mixing hSAA1 and hFbg or hCitFbg had peaks, corresponding to both

hSAA1 as well as either hFbg or hCitFbg, indicating that a large part of the SAA might not be bound to Fbgs. However, the inclusion of hSAA1 for both samples was demonstrated in the western blot analysis of the fraction corresponding to the peaks of hFbg and hCitFbg. Furthermore, the data revealed that the hCitFbg fraction contained more hSAA1 than hFbg (Fig. 6b and Fig. S6b). Next, we examined the hydrodynamic properties of hFbg and hCitFbg using dynamic light

scattering (DLS) in the absence and presence of hSAA1 (Fig. 6c). In the absence of hSAA1, hFbg, and hCitFbg exhibited relatively mono-dispersed correlations, from which average hydrodynamic diameters of 28.8 nm and 32.8 nm, respectively, could be calculated. In the presence of hSAA1, hFbg, and hCitFbg had correlation curves distinct from those in the absence of hSAA1, corresponding to hydrodynamic diameters of 22.6 and 23.8 nm, respectively. The same relationship ($d_\mathrm{H}^\mathrm{CitFbg} > d_\mathrm{H}^\mathrm{Fbg}$) was maintained by the data detected in the absence of hSAA1. Nonetheless, the unexpected contraction of both hFbg and hCitFbg suggested that the linear rod-like structure of Fbgs might be altered by the binding of hSAA1. Additionally, the association of hSAA1 with hFbg or hCitFbg was unequivocally demonstrated by the data. SAA is well known for causing SAA amyloidosis by forming insoluble amyloid fibrils resistant to degradation[34], and the alpha chain of Fbg contributes to the amyloidosis[35,36]. In spite of this, the correlation curves after adding SAA1 exhibited no apparent aggregations, that could cause strong light scattering and slow correlation time.

Next, we decided to investigate whether SAA-CitFbg aggregation could be established in a culture medium with human lung ECs. Human lung ECs were cultured in the presence of purified Fbg or CitFbg with or without recombinant SAA1 for 1 week, and SAA1 and citrullinated proteins suspended in the culture media were detected by immunostainings. Our results revealed that EC incubation with CitFbg and SAA1 produced more aggregations than other conditions (Fig. 6d). Thus, it is highly likely that SAAs expressed on ECs combined with CitFbg, which was citrullinated by PAD-proteins derived from ECs in the extracellular sphere, to form an aggregation complex. Moreover, our DLS data implies that SAA1 enhanced the formation of CitFbg aggregations. The addition of CitFbg after SAA1 solo-incubation resulted in a rightward shift of the correlation curve, suggesting that SAA1 pre-incubation prior to co-incubation with CitFbg generated larger complexes than simple SAA1 and CitFbg co-incubation (Fig. S6c). Given the fact that SAA1 binds with CitFbg, pre-incubation of SAA1 alone provided a seed to form a large aggregation. Finally, we examined the enzymatic degradation of Fbgs by plasmin. It has been known that Fbg is cleaved by plasmin, and its citrullination resists the cleavage reaction[37]. We tracked the degradation of SAA by plasmin (Fig. S6d) and followed the degradation process of hFbg and hCitFbg in the presence and absence of hSAA1. As shown in Fig. 6e, no significant difference in the degradation of Fbg and CitFbg was detected in the current condition. Moreover, the degradation of hFbg and hCitFbg was not adversely affected by the addition of hSAA1. Thus, the citrullinated protein complex could be eliminated in the pre/post-metastatic microenvironment.

We examined the attachment and aggregation of human breast cancer cell lines, such as MDAMB231 and MCF7 cells, to the hSAA-CitFbg protein complex in vitro to confirm the biological importance of the SAA-CitFbg complex. Tumor cells first need to perform physical trapping on an endothelial bed before its extravasation in tumor metastasis. As observed in the in vivo data (Fig. 3), the CitFbg-coated plates bound more stimulated tumor cells than the Fbg-coated plates (Fig. S7a), and the addition of SAA to CitFbg further enhanced tumor cell attachments (Fig. S7b, c). Furthermore, it was noteworthy that the SAA-CitFbg complex boosted the aggregation (Fig. 7a, b). Therefore, microenvironments with SAA-CitFbg accumulation could facilitate tumor cell metastasis.

So far, a few valuable reports have indicated that the N-terminal and C-terminal peptides of CitFbg were helpful in the diagnosis of RA[38], indicating that these residues on the surface of CitFbg protein were exposed. We prepared four peptides for the N-terminal and C-terminal of the alpha chain of Fbg and CitFbg (referred to as FGA-1, FGA-C-2, FGA-3, and FGA-C-4, respectively) in our attachment assay. The amino acid sequences of FGA-1 and FGA-3 are taken from the N- and C-terminals of Fbg, respectively. All arginine residues were converted to citrulline in FGA-C-2 and -C-4. Interestingly, FGA-1 was found to

obscure the thrombin cleavage site, so its citrullinated peptide (FGA-C-2) is expected to work as an inhibitor. These peptides were added as competitors in the cell attachment assay. Our screening found that MDAMB231 cell attachment was strongly suppressed by the excessive application of the N-terminal and C-terminal CitFbg peptides (Fig. 7c). This competition assay showed that FGA-C-2 prevented MCF7 and MDAMB231 attachment (Fig. 7d). Finally, the in vivo application of the peptide reduced metastatic formation in the lungs in which SAA-CitFbg deposition was preconditioned (Fig. 7e).

In summary, our SAA-humanized mouse satisfactory exhibited the pre-metastatic niche in the mouse model system. Additionally, the details of the SAAs-CitFbg complex were unveiled by this model, and it was seen to have a high affinity for human tumor cells (Fig. 7f and Fig. S7d). SAA-induced PADs stimulated CitFbg conversion, and this led to the formation of SAAs-CitFbg aggregation in a few sites. In this prospective study, a pre-metastatic site was detected by a specific antibody against CitFbg protein, and a potential blockade by a specific CitFbg peptide was shown. These results may contribute to the detection and inhibition of metastasis on the basis of detection of the early- metastatic region in cancer patients.

## Discussion

The concept of a pre-metastatic microenvironment was proposed more than a decade ago[8,9]. A strand of mouse model studies revealed that distant primary tumors found an environment that was suitable for metastasis in the pulmonary vessels. Remarkably, this environment existed in distinctly hyperpermeable regions, and contained various types of BMDC stressed by signaling molecules from the primary tumor[5,16]. Further studies revealed that every organ (e.g., liver, lung, and bone) has characteristic molecules that play an important part in metastasis. Few earlier studies have demonstrated that gene knockout therapy or antibody therapy aiming to block the specific molecule effectively suppresses metastasis[5,7]. In our previous studies, we have presented that Fbg depositions in autopsy samples taken from cancer patients are very similar to those found in pre-metastatic mouse lungs[16]. These data imply that eliminating such Fbg depositions would be a very efficient treatment in the pre- or early-metastatic phases.

We showed one prospective study that used humanized mice, and some other retrospective studies that used autopsy samples. These data would be beneficial for the preclinical model[39]. Our transcriptome-wide screening using autopsy samples has identified SAAs and HMGB1 as potential markers to spot the pre-metastatic region. Mouse study publications have concluded that these molecules are peculiar to the pre-metastatic phase[5,11,40]. In our previous study, we reported that pulmonary vessels of autopsy samples contained S100A8/9, as expected in cancer patients[16]. S100A8/9 are upstream molecules of SAA3 in mice, and its human ortholog is SAA1. Thus, these facts indicated that mouse study data could be applied to human research to find out more about the pre-metastatic site. Additionally, the significance of hCitFbg depositions in the pre-metastatic phase was understood in this study. The detection of hCitFbg in immunohistochemical staining was accomplished for the first time, because the specificity of our antibody against hCitFbg was better than that of the other available antibodies. This study also demonstrated that the metastatic microenvironment was regulated by hCitFbg. In humanized mice, metastatic tumor cells were recruited by the SAAs-hCitFbg protein complex. Considering these facts, it is worth investigating SAAs-hCitFbg in a prospective study.

SAA and Fbg are well-known sources of amyloidosis[35,36]. The presence of these proteins in high concentrations may present a pathogenic risk factor. Once an amyloid fibril is formed, it becomes hard to eliminate since it is insoluble and resistant to proteolytic degradation[35,36]. Therefore, the status of SAAs-Fbg is an important aspect of pathogenesis. The treatment would become more complicated if the SAA-CitFbg complex were to have an amyloid fibril-like

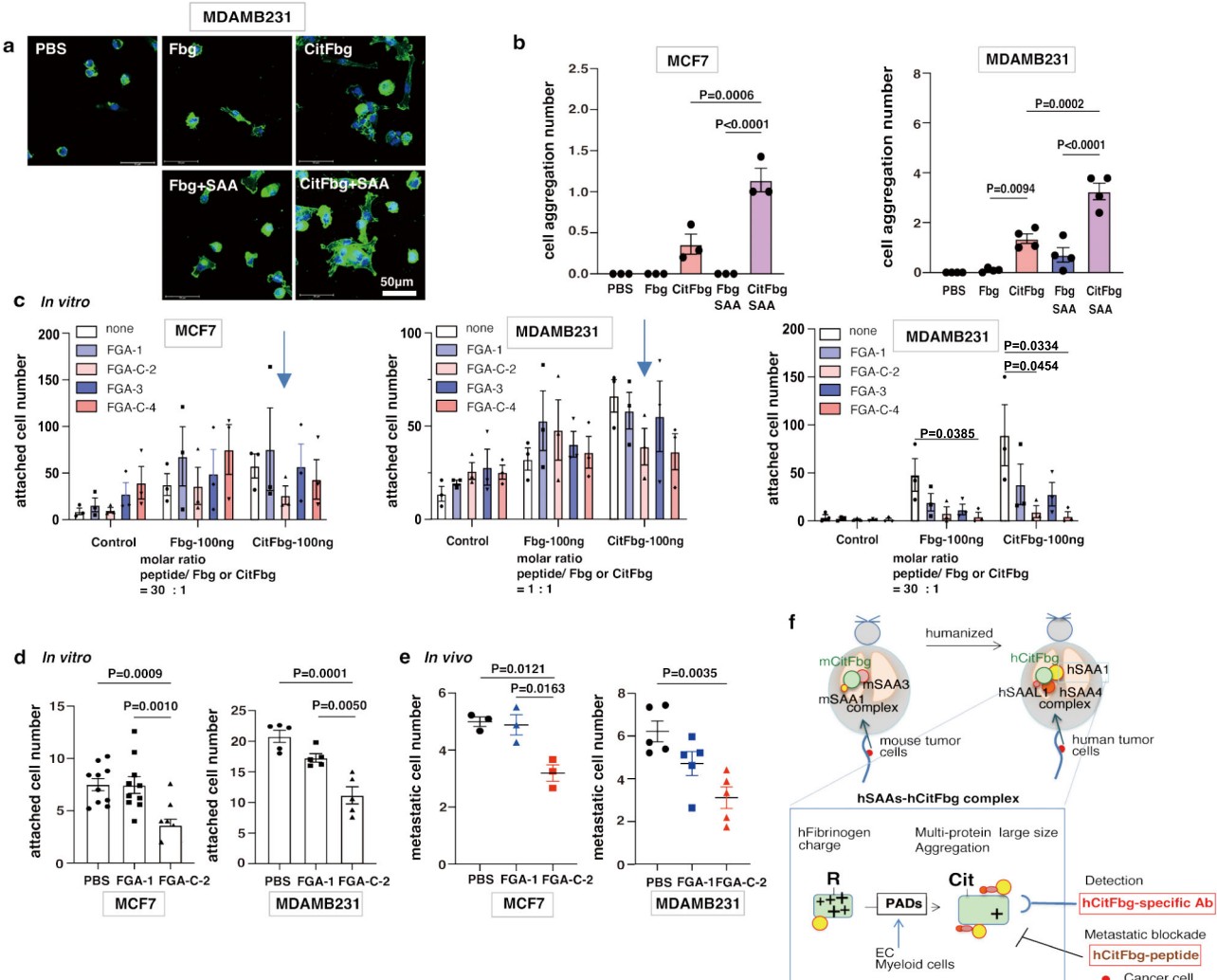

**Fig. 7 | Inhibition of metastagenesis by specific CitFbg peptide. a, b** Human tumor cell aggregation assay for measuring Fbg, CitFbg, SAA1-Fbg, and SAA1-CitFbg proteins. Representative photo of aggregated MDAMB231 cells with SAA1-CitFbg protein complex (Scale bars, 50 μm in (**a**). The cell aggregation number (five or more cells in aggregation) of MCF7 and MDAMB231 cells cultured on various protein complex-coating plates for 3 days (**b**). ($n = 3$ for MCF7 and $n = 4$ for MDAMB231 cells). One-way ANOVA with Bonferroni correction. Data were mean ± SEM. **c** Screening of CitFbg peptides in inhibition assay against the tumor cell attachment to Fbg and CitFbg ($n = 3$). FGA-1, FGA-C(citrulline)-2, FGA-3, and FGA-C-4 were 15 amino acid length peptides derived from the N-terminal or C-terminal of the Fbg and CitFbg protein sequences, respectively. One-way ANOVA with Bonferroni correction. Data were mean ± SEM. **d** Competitive inhibition assay in vitro

against the tumor cell attachment to Fbg and CitFbg using N-terminal peptides of Fbg and CitFbg. MCF7 cells ($n = 10$) and MDAMB231 cells ($n = 5$) were shown after applying the appropriate peptides. Data were mean ± SEM. Two independent experiments. **e** In vivo metastasis inhibition assay using FGA-1 and FGA-C−2 peptides. The number of metastatic tumor cells in the lung after application of the CitFbg protein, followed by the peptides. ($n = 3$ C.B17/Icr-scidJcl for MCF7 and $n = 5$ C.B17/Icr-scidJcl for MDAMB231). One-way ANOVA with Bonferroni correction. Data were mean ± SEM. **f** Model to detect and suppress pre-/early-metastatic sites in cancer patients. A retrospective study of cancer patients indicated that hSAAs-hCitFbg is a presumptive pre-metastatic molecule. A prospective study demonstrated that the formation of hSAAs-hCitFbg recruits cancer cells to facilitate metastasis.

structure or an irreversible aggregation. However, the DLS analysis indicated that the SAAs-Fbg and SAAs-hCitFbg were soluble protein complexes with no amyloid fibril formation. This structural analysis was supported by the plasmin test. SAAs-hCitFbg and SAAs-Fbg were found to be degraded at the same rate.

In this study, we have tried to elucidate molecular mechanisms of Fbg citrullination and the formation of a limited number of the SAAs-Fbg hotspots in the pre-metastatic lungs. There are three important points to consider in this context: The type of the cell responsible for citrullination, the place of the citrullination reaction (intracellular or extracellular), and the reaction scheme to form SAA-CitFbg aggregations. First, we identified that lung ECs and CD11b+ myeloid cells mediated Fbg citrullination in a PAD4-dependent manner when they are stimulated by the primary tumor. It has been reported that deletion

of PAD4 in neutrophils or pharmacologic inhibition of PAD4 reduced both primary tumor growth and lung metastases[41,42]. Thus, it is likely that PAD4 is involved in the pre-metastatic citrullination in the lungs. PAD2 was reported as an oncogene in prostate cancer progression[43]. Our in vitro knockdown experiments showed that PAD2 siRNA transfection, as well as PAD4 siRNA transfection, reduced Fbg citrullination by ECs. Therefore, although PAD4 is the primary molecule for Fbg citrullination, the effect of PAD2 cannot be ruled out at this point. Second, we demonstrated that hSAA1 stimulated the production of PADs in lung organ culture, and that SAA1 might be an impetus for the formation of Fbg aggregation in lung EC culture medium. This is likely because extracellular PADs from neutrophils have been reported to citrullinate Fbg[22]. Third, we observed SAA-CitFbg aggregations in the medium with ECs after 1 week when cultured. Large-size aggregations

were observed when the culture was initiated in the presence of both, SAA1 and CitFbg. Both components were necessary for the aggregation. The effectiveness of SAA1 seeding in the CitFbg aggregation was also shown in our DLS assay. The presence of CitFbg accelerated the aggregation formation. Even in the case of the external addition of CitFbg, PADs may function as an aggregation promoter. Some research groups identified citrullination sites in Fbg. The analysis reported that 22–27 arginine residues of the Aα chain were citrullinated in the in vitro optimized condition, but some arginine residues were citrullinated to a greater extent than others[44,45], indicating that all target arginine residues were not equally citrullinated even though they can be recognized as a substrate. These data imply that CitFbg in vivo might be composed of a mixture of citrullinated and non-citrullinated residues. Our artificial CitFbg might get further citrullinated by PAD secreted from ECs and myeloid cells during the incubation period. To form large-size CitFbg aggregation, the aggregation complex must get reduce the CitFbg molecules from the vicinity. This indicates that the formation of large aggregations results in reduced CitFbg concentration in the vicinity. As a result, the CitFbg levels reduce when an aggregation is created, thereby, reducing the chances of formation of more aggregates. This is proved by the fact that a limited number of CitFbg aggregations are observed in our mouse model system.

Citrullination of proteins has been well recognized in the pathogenesis of cancer and RA[46]. This enzymatic reaction may affect the protein structure and protein–protein interactions, because citrullination alters the positively charged guanidino group on the protein surface[47]. We have successfully demonstrated that hCitFbg bound more with SAAs than with hFbg, and this difference was reflected in their hydrodynamic radii. There is no doubt that variations in any microenvironment are triggered by post-translational modification of the extracellular sphere. This study was the first one to reveal the role of the citrullination of Fbg in the creation of a pre-metastatic niche. The fact that Fbg is citrullinated in the pre-metastatic lung enables us to narrow down the metastatic regions. Furthermore, the detection of SAAs and Fbg citrullination positive area would increase the accuracy of the prediction of future metastasis. Our hCitFbg-specific antibody can provide a new approach for therapeutic intervention of pre- or early-metastasis.

## Methods
### Fibrinogen purification and citrullination
Human fibrinogen protein (hFbg, Sigma-Aldrich) diluted in 50 mM Hepes, pH 7.4, was loaded on an anion exchange column (Hitrap Q-Sepharose HP, 5 mL, Cytiva, Tokyo, Japan). The column was washed with the dilution buffer (50 mM Hepes, pH 7.4) and eluted with a linear gradient (50 mM Hepes, pH 7.4- 50 mM Hepes, pH 7.4, 1 M NaCl). The peak fraction (0.2 M NaCl–0.3 M NaCl) was collected and concentrated by using a centrifugal concentrator (Sartorius). Purified hFbg was further dialyzed against the dilution buffer. A part of purified hFbg was citrullinated by recombinant human PAD2 (Cayman Chemical, Ann Arbor, MI) in a deimination buffer (50 mM Hepes, pH 7.7, 10 mM CaCl$_2$, 5 mM DTT) for 2 h at 37 °C. Citrullinated fibrinogen (hCitFbg) was purified using Hitrap Q-Sepharose HP, and the conditions were similar to those used to purify hFbg. The peak fraction (0.25 M- 0.35 M NaCl) of hCitFbg was collected, concentrated, and dialyzed against the dilution buffer. Purified hFbg and hCitFbg were concentrated and stored in a deep freezer (−80 °C).

### Human samples
Tissue samples of human lungs taken from patients for autopsy were acquired after obtaining informed consent, followed by opt-out methods. The quality of the lung tissues was confirmed by using an anti-VE-cadherin antibody. The normal control group included patients with cardiomyopathy, arrhythmia, aortic dissection, pulmonary embolism, aneurysm, cardiac infarction, cerebral infarction,

cerebral hemorrhage, and degenerative neuron disorders. For the cancer group, the non-metastatic and metastatic lung lobes from patients with various primary tumors, including pancreatic cancer, hepatocellular carcinoma, gastric cancer, cholangiocellular carcinoma, prostate cancer, gallbladder cancer, breast cancer, colon cancer, and esophageal cancer, were used (noncancer, $N = 26$; cancer, $N = 67$. Sample list is depicted in Supplementary Table 2). Regarding IHC analysis, the median age of the patients was 70 years (range, 33–91 years), and the gender ratio was male (51%) and female (49%). Samples were microscopically examined to eliminate cases that had abnormal changes, such as micrometastasis and severe inflammation. Joint samples from RA and non-RA patients ($N = 2$) were used with informed consent. Sex and gender were not considered in this study design. Sex and gender analysis was not carried out because it is not necessary to analyse tumor metastasis. All samples were analyzed under the approval of the Ethics Committee of the Shinshu University School of Medicine (No. 5326 and 5338) and the Institutional Review Board of the Tokyo Women's Medical University (No. 2533). The investigation was conducted in compliance with the Declaration of Helsinki.

### Laser microdissection
Human lung specimens were stained with HE and anti-Fbg antibodies. The small pulmonary blood vessels were prepared under the reference of Fbg markers in the sections derived from noncancer and cancer patients. Microdissection was conducted using PALM MicroBeam IV (Zeiss) with PALM Robo 4.6 (Zeiss).

### Cell culture
Mouse LLC (RCB0558) cells were obtained from Riken BRC Cell Bank (Tsukuba, Japan), and a highly metastatic subline, 3LL (JCRB1348), was purchased from JCRB Cell Bank (Osaka, Japan). E0771, equivalent to ATCC CRL-3461, was provided by the original source[48]. The cells were maintained in Dulbecco's modified Eagle's medium (DMEM) (Gibco) supplemented with 10% FBS, 100 units/mL penicillin G sodium, and 100 µg/mL streptomycin sulfate. For human tumor cell lines, human triple-negative breast cancer cell lines MDAMB231 (HTB-26) and human estrogen receptor-positive luminal breast cancer cell line MCF7 (HTB-22) were obtained from the ATCC. Human cells were maintained in RPMI-1640 (Gibco) supplemented with 10% FBS. Normal human Fbg-producing CHO cells were established by co-transfection with human Fbg (A)α-, (B)β-, and γ-chain-cDNA containing plasmid, and were cultured either in DMEM Ham's nutrient mixture F12 (DMEM/F12), supplemented with 5% bovine calf serum (HyClone Laboratories, Logan, UT, USA)/5% Nu-serum (BD Biosciences, Bedford, MA, USA).

### Animals
We generated mouse SAAs-cluster KO mice (Trans Genic Inc., Kumamoto, Japan) and human SAAs-cluster transgenic mice (Institute of Immunology Co. Ltd., Tochigi, Japan). The mouse *Saa* cluster genes (NCBI Gene ID: 111345, -70 kbp) were deleted using the CRISPR/Cas9 genome-editing strategy. The guide RNA (gRNA) was designed using the optimized CRISPR design tool (Massachusetts Institute of Technology, Zhang Lab). The guide sequences for the knocking strategy of the *loxP* sequence were 5'-GTCCCCCCGACCCGTATGCAAGG-3' and 5'-AGCGTTTATTTGACTCGTACAGG-3' for the *Saa1* and *Saa2* sites, respectively. Then, we obtained the *Saa* cluster gene deletion ES clones (C57BL/6) which had been confirmed by direct sequencing. Regarding human *SAAs*-cluster transgenic mice, the RecBAC vector, BAC transgenic construct harboring human *SAA* cluster genes were constructed using human BAC genome clones RP11-81D23 and RP11-796A15 with Red/ET reaction (GeneBridges, Germany) was microinjected. After setting up the specific probes for *hSaa1, hSaa2, hSaa3P, hSaa4*, and *hSaal1* in Southern blot analysis, we screened 106 founder mice (shown in Fig. S1e), and six lines (C57BL/6J) were obtained as *hSAAs* cluster transgenic mice. mSAAs-/- mice and three lines of hSAAs-

Tg mice were bred to obtain hSAAs-Tg/mSAAs-/- mice. The hSAAs-Tg/mSAAs-/- mice were bred with Rag1 KO mice[49] to generate hSAAs-Tg/mSAAs-/- /Rag1-/- mice.

## Mouse metastasis models

C57BL/6, NOD/ShiJic-scidJcl, and C.B17/Icr-scidJcl mice were purchased from Clea Japan (Tokyo, Japan) or SLC (Shizuoka, Japan). Mice were housed in a specific pathogen-free condition in a controlled environment with a constant temperature (22 °C) and humidity (50%) under a 12 h/12 h light/dark cycle. All animal procedures were performed according to the guidelines of the Animal Research Committee of Shinshu University and received ethical approval from the Committee. In this study, 6- to 10-week-old female or male mice were used.

LLC or E0771 cells were implanted subcutaneously (s.c.) or via mammary fat pad implantation, in which $1 \times 10^6$ tumor cells were implanted and maintained for 12–14 days to generate tumor-bearing mice. The maximal tumor size in this study is 10-mm. This follows the criterion in 'Guidelines for Endpoints in Animal Study Proposals' released by NIH. LLC cells were implanted in 6–10-week-old male mice and E0771 cells were in 6–10-week-old female mice, respectively. For the tumor cell homing assays, $1 \times 10^4$ fluorescent dye (PKH26, Sigma-Aldrich, St. Louis, MO, USA)-labeled cells were suspended in 100 μL of PBS and injected intravenously. Lungs were excised 24–48 h after the injection. Lung tissue fragments (2 mm in diameter) were randomly chosen and embedded in an Optimal Cutting Temperature compound (Sakura Finetek Japan, Tokyo, Japan) before freezing at −80 °C. Sections (10 μm thickness) were obtained from the frozen samples using a CM3050S cryostat (Leica Biosystems, Nussloch, Germany), and three 10-μm sections per fragment were examined under confocal (Leica TCS SP8, Heidelberg, Germany) or fluorescent microscopes (BZ-9000, Keyence, Osaka, Japan) with BZ-II viewer 1.40 (Keyence). The labeled tumor cell counts were normalized according to the total tissue surface area.

## Fbg or CitFbg treatment

Purified Fbg or CitFbg (50 ng/animal) was diluted in 200 μL PBS and injected through the tail vein, 1 h before the administration of labeled tumor cell injection, in order to determine the effects of Fbg and CitFbg on tumor metastasis. In the case of the Fbg peptide treatment, 50 ng of the peptide was injected 1 h after the Fbg or CitFbg injection. In this case, the labeled cell injection was carried out 1 h after the peptide injection.

## Mouse macroscopic metastasis assays

One hour after the CitFbg injection, $1 \times 10^6$ cells of LLC, E0771, MCF7, or MDAMB231 cells suspended in 100 μL of PBS were additionally injected into the animals. Three weeks later, the lungs were excised and fixed in 4% PFA to analyze the metastatic nodules.

## Analysis of mouse peripheral blood cells

Whole blood samples were obtained from the anaesthetized mice using a Goldenrod animal lancet (5 mm point size, Braintree Scientific Inc.). The blood sample was collected from the back of the jaw and were transferred into EDTA-coated collection tubes. For fluorescence-activated cell sorting (FACS) analysis, whole blood samples were treated with RBC lysis buffer (Sigma-Aldrich) for 10 min at rt. Then, an equal amount of PBS was added before centrifuging the samples at 300×g at rt. The resulting cell pellet was washed with PBS-1%BSA and resuspended in PBS-1%BSA containing Fc-Block (1/200; Clone 2.4G2; BD Pharmingen). The antibody was incubated at 4 °C for 30 min. Antibodies used for cell staining are as follows: anti-NK1.1-BV421 (1:200, 108741, BioLegend), anti-B220-PE/Cy7 (1:200, 103222, BioLegend), anti-CD3-FITC (1:200, 100204, Biolegend), anti-CD11b-APC (1:200, 101211, Biolegend), anti-CD11c-PE (1:200, 117307, BioLegend).

## Isolation of cells from mouse lungs

Mouse lungs were digested with 10 mg of collagenase (Fujifilm Wako), 10 mg of dispase (Gibco), and 10 μg of DNase (Worthington Biochemical, Lakewood, NJ) in 10 mL of serum-free DMEM at 37 °C for 45 min. The cells were resuspended in PBS-1% BSA and sequentially filtered through 100- and 40-μm nylon filters (Falcon). After treatment with red blood cell lysis buffer (Sigma-Aldrich), cells were resuspended in PBS-1% BSA and loaded onto a cell sorter (FACSAria III, BD Biosciences). For FACS sorting, cells were labeled with the following antibodies: anti-CD144-PE/Cy7 (1:200, 138015, BioLegend), anti-CD11b-APC (1:200, 101211, BioLegend), and anti-F4/80-FITC (1:400, 123107, BioLegend). FACS data were analyzed using FACS Diva 8.0.2 (BD) and Flowjo v7.6 (Flowjo LLC).

## Fbg plate coating

Fbg (Human, Sigma-Aldrich) was diluted to obtain working concentrations (0, 10, 75, 100, 250, and 1000 μg/mL) in PBS. A cover glass was placed in every plate of the six-well plate and incubated at rt for 1 h or at 4 °C overnight. The following day, the plates were washed three times with PBS. Sorted cells were seeded in the well and cultured in serum-free DMEM. After 48 h, the culture medium and cells were removed, and coated Fbg was retrieved in RIPA buffer (50 mM Hepes, pH 7.7, 150 mM NaCl, 0.5% Triton X-100, 0.5% sodium cholate) with protease inhibitor cocktail. Citrullinated Fbg was immunoprecipitated by anti-citrulline antibody (1:50, MA5-27573, Thermo) and protein G beads. Beads were washed five times with PBS-T and treated with SDS sample buffer for western blot analysis.

## Immunocytochemistry (ICC) for plate

Cells were fixed with methanol, permeabilized in 0.1% Triton X-100, and blocked with Blocking One (Nacalai Tesque, Kyoto, Japan) for 1 h. Anti-citrulline antibody (1:200, MA5-27573, Thermo) probing was conducted overnight at 4 °C. After washing with PBS, Alexa-594-conjugated secondary antibody (1:400, A21203, Thermo) was used to detect citrullinated proteins, and DAPI (BioLegend) was used to stain the nuclei. Images were acquired using a Cell Observer microscope (Zeiss), and the cell area and the number of apoptotic cells per field were analyzed with Photoshop CS6 (Adobe).

## Electroporation of PAD2 or PAD4 siRNA to mouse cells

The EC and CD11b cells in the lungs from the primary tumor-bearing mice were FACS-purified and seeded on collagen-coated dishes for culturing in EBM-2 basal medium with EGM-2 MV supplements (Lonza) and RPMI-1640 (Gibco) supplemented with 10% FBS, respectively. After 48 h incubation, siRNA for PAD2 or PAD4 (ON-TARGET plus, Horizon Discovery Ltd.) was introduced into the cells by electroporation and then the cells were seeded on an 8-well chamber slide coated with purified fibrinogen (100 ng/ well). The cells were immunostained using anti-PAD2 (1:200, 12110-1-AP, Proteintech) and anti-PAD4 (1:200, 17373-1-AP, Proteintech).

## Immunofluorescence staining

Anti-mouse SAA1/2 (1:100, ab199030, Abcam), anti-citrulline (1:200, ab100932, Abcam), anti-citrulline (1:200, MA5-27573, Thermo) anti-CCP Polyclonal (1:200, bs-1053R, Bioss), anti-mouse CD45 (1:200, 103101, BioLegend), anti-mouse CD11b (1:200, 101202, BioLegend), and rat anti-mouse MECA32 (1:200, 550563, BD Pharmingen), anti-Fbg (1:200, A0080, Dako), anti-PAD2 (1:300, Proteintech), anti-PAD4 (1:300, Proteintech), and anti-citrullinated Fbg clone A-s1 (1:100, produced in our laboratory), were used for immunostaining the frozen mice lung tissue sections. We also used anti-mouse SAA3 antibody (1:100, produced in our laboratory) and Alexa Fluor secondary antibodies (1:400, A21208, A21207, A21447, A21202, A21203, A21206, Thermo, and ab150111, ab150075, ab150156, ab150155, Abcam) for signal visualization during confocal microscopy. For human transgenic

SAA staining, we used three antibodies for human SAA1/2: (1) ab200584 (1:100, Abcam), (2) ab207445 (1:100, Abcam), and (3) ab687 (1:100, Abcam). For human SAA4 staining detection, we used an anti-SAA4 (1:100, orb541488, Biorbyt) antibody. Immunofluorescent images were first exported by Leica Application Suite X 3 (Leica Microsystem). The exported images were then analyzed and quantified using Photoshop CS6, ImageJ 1.52 (NIH), or BZ-II analyzer 1.42 (Keyence). The immunostained signals are shown as numbers of pixels normalized to the signals of DAPI. Photoshop was used to calculate the distance or area. The labeled tumor cells were taken at the center of the immunofluorescence-stained sections. The proportion was converted according to the Photoshop scale and the scale in the actual photos (20 μm = 14.8 mm). Then, the pixel of the protein in the frame was calculated after frame processing.

### Immunoprecipitation and western blot analysis

Lung tissues were lysed with lysis buffer (50 mM Hepes, pH 7.4, 0.5% Triton X-100, 0.5% sodium cholate, 150 mM NaCl) supplemented with a protease inhibitor mixture. Protein concentrations were determined using a BCA assay (Thermo). Equal protein concentrations were diluted in SDS-PAGE loading buffer, boiled at 95 °C for 3 min, and then loaded on SuperSep 10–20% gels (Fujifilm Wako). Proteins were transferred onto 0.2 μm PVDF membrane (Fujifilm Wako) or 0.45 μm PVDF membrane (Merck Millipore). The membrane was blocked for 1 h at room temperature in Blocking One (Nacalai Tesque) and then incubated overnight at 4 °C with the indicated antibodies diluted in reagent A (HIKARI, Nacalai Tesque). We used four antibodies for citrulline: (1) ab100932 (1:1000, Abcam), (2) MA5-27573 (1:1000, Thermo), (3) ab240908 (1:1000, Abcam), and (4) bs-1053R (1:1000, Bioss Antibodies). For Fbg detection, we used anti-Fbg (1:1000, A0800, Dako) and anti-Fbg (1:1000, ab119948, Abcam) antibodies. For mouse SAA detection, we also used anti-mouse SAA3 antibody, as described above (1:1000), and anti-SAA antibodies: (1) anti-SAA1 + 2 (1:1000, ab199030, Abcam), (2) anti-SAA4 (1:1000, orb541488, Biorbyt), and (3) anti-SAAL1 (1:1000, orb221911, Biorbyt). For human transgenic SAA detection, we used two antibodies for the human SAA1/2 blot: (1) ab207445 (1:1000, Abcam) and (2) ab687 (1:1000, Abcam). For the human SAAL1 blot, we used an anti-SAAL1 (1:1000, HPA039003, Sigma-Aldrich) antibody. Anti-SAA4 (1:1000, orb541488, Biorbyt) antibody was used in the western blot. The membrane was washed three times with TBST for 10 min and incubated for 1 h at room temperature with the corresponding secondary HRP conjugate antibody (1:10000, NA931, NA934, Cytiva) diluted in reagent B (HIKARI, Nacalai Tesque). The membrane was washed three times with TBST for 10 min and then developed using an ECL solution (Fujifilm Wako). All uncropped blots are displayed in Fig. S8.

### Cell attachment assay

Purified human Fbg and CitFbg (1 mL, 100 ng/mL) were coated on a 24-well untreated culture plate overnight at 4 °C and washed with PBS three times. Fbg peptide was added (400 μL) in the wells, and blocking was done for 30 min at 37 °C in an incubator. The peptide concentration was 0.3 or 10 ng/ml. Then, 300 μL of the cell suspension ($1 \times 10^4$ cells /mL) was added to each well. To avoid nonspecific cell adhesion, PBS-0.75% BSA for MCF7 cells and 20% low glucose-DMEM-80% HBSS for MDAMB231 cells were used as cell suspension solutions. After incubation for 1 h, the wells were washed five times with PBS, and the cells were stained with Diff-Quik (Sysmex). Finally, the number of cells was counted.

### Cell aggregation assay

Purified human Fbg, CitFbg, and hSAA1 (1 mL, 50 ng/mL) were coated on cover glasses overnight at 4 °C and washed with PBS three times. Then, the Fbg(hSAA1) and CitFbg(hSAA1) coated cover glasses were placed in the non-coated well. The PBS-coated cover glass was used as

a control. Two mL of cell suspension ($1 \times 10^6$ cells /mL) was seeded into each well. PBS-0.75%BSA for MCF7 cells and 20% low glucose-DMEM-80% HBSS for MDAMB231 cells were used as cell suspension solutions for preventing nonspecific cell adhesion. Cells were cultured for 8 h to 3 days, and five images from the chamber were captured and viewed under an inverted microscope (CK40 Olympus, Tokyo, Japan) with a USB camera for Microscope Viewer Software v070817 (Shodensha, Osaka, Japan). On day 3, the cells were stained with phalloidin-iFluor 488 conjugate (1:1000, 20549, Cayman Chemical) and DAPI.

### Protein aggregation assay

Primary human lung microvascular cells (Applied Cell Biology Research Institute, ACBRI 468) were seeded in EBM-2 Microvascular Endothelial Cell Growth Medium-2 SingleQuots Supplements and Growth Factors (Lonza) on a collagen-coated 8-well chamber slide (Watson). Various combinations of Fbg, CitFbg, and SAA1 (50 ng/mL, 200 μL) were added to the culture medium twice in a 7 day-culture. The conditioned media were centrifuged at 20,000×$g$ and stained with anti-citrulline (0.5 mg/mL, bs-1053R) and anti-SAA1 antibodies (0.5 mg/mL, ab207445) followed by Alexa Flour 488-conjugated secondary antibody (1:400, A21206, Thermo). Protein aggregations were gently washed, and all aggregations were dried up on the slide to be further analyzed under a confocal microscope (Leica TCS SP8, Heidelberg, Germany).

### Organ culture

Lungs derived from C57BL/6 wild-type mice were cut into 2 mm cubes and cultured with various factors such as mSAA3, mS100A8, hSAA1 (300-53, Peprotech), mCCL2 (479-JE, R&D systems), mSDF1 (ab240836, Abcam), and mVEGF (493-MV, R&D systems) in DMEM-1% FCS for 72 h. mSAA3 and mS100A8 were prepared in our lab using *E. coli* overexpression system. Concentrations of factors were 100 ng/mL for mSAA3, hSAA1, mS100A8, mCCL2, and mVEGF and 50 ng/mL for SDF1. The specimens were embedded in OCT compound and cut, then slides were stained using anti-PAD2 (0.5 mg/mL, Proteintech) and anti-PAD4 (0.5 mg/mL, Proteintech) antibodies and control IgG (0.5 mg/mL, 3900 S, Cell Signaling Technology). Signals were examined by a fluorescent microscope (BZ-9000, Keyence, Osaka, Japan).

### Patients' immunohistochemistry assays

Immunohistochemistry staining was performed according to the standard method, including a deparaffinization and rehydration step. Endogenous catalase was quenched with 3% hydrogen peroxide (methanol: 30% hydrogen peroxide = 9:1) for 10 min. Tissue sections were pretreated with heat-induced antigen retrieval using 0.01 M citrate, pH 6.0, for CitFbg single staining and Daka target retrieval solution, pH 9 (S2367, Agilent) for double staining, including VE-cadherin, hSAA, or Fbg, for 15 min in a microwave.

**Single staining.** Samples were stained with primary mouse anti-CitFbg antibody (1:100, clone A-s1) overnight at 4 °C in a humid chamber, followed by 30 min of secondary antibody incubation (1:1000, NA931, Cytiva) at room temperature in a humid chamber. Samples were developed using DAB. Reactions were terminated by washing with water. The samples were stained with hematoxylin for 15 s and then washed under running water for 3 min. Finally, the samples were dehydrated and sealed with neutral resin.

**Double staining.** Nonspecific binding was blocked with Blocking One (Nacalai Tesque) for 30 min at room temperature. A sequential double staining procedure[50] was initiated with CitFbg, and was visualized by BrightVision HRP-conjugated anti-mouse IgG polymer (DPVM-HRP, Immunologic) and Bright DAB (BS04, ImmunoLogic). The DAB reaction product effectively sheltered the first set of antibodies. Then, the process was continued with VE-cadherin (1:200, sc6458, Santa Cruz)

and anti-human SAA1/2 (1:1000, ab207445, Abcam) and anti-human Fbg (1:1000, A0800, Dako), and was visualized by BrightVision alkaline phosphatase (AP-)conjugated anti-rabbit IgG polymer (DPVR-AP, ImmunoLogic) and Vector Blue (SK-5300, Vector Laboratories; Burlingame, CA). The slides were counterstained, washed with tap water, and coverslipped with PathodMount (Fujifilm Wako) after dehydration with Clear-Rite 3™ (Richard-Allen Scientific LLC, USA). Images were acquired with Vectra 3 (PerkinElmer), Phenochart 1.1.0, and inForm 2.4.9 (Akoya Bioscience).

## Multiplex IHC data analysis

Immunostainings of lung samples from noncancer and cancer patients were analyzed using the following steps to quantify Fbg, CitFbg, and SAA1/2 in the small pulmonary vessels. All slides were scanned at low-resolution mode (4x magnification) using Vectra 3 with a Phenochart. The images containing blood vessels were further scanned in high-resolution mode (0.5 μm per pixel, 20x magnification). inForm Tissue Analysis Software was used to detect Fbg, CitFbg, or SAA1/2 in high-resolution images. This software is equipped with an artificial intelligence type of staining pattern recognition algorithm. Therefore, once the software was programmed, it allowed us to automatically measure protein signals around the blood vessels from the images. Finally, the signals of Fbg, CitFbg, or SAA1/2 in the defined regions were analyzed. At least 25 areas were selected for each multi-staining pattern in each patient.

## Gel filtration column chromatography

The sample was loaded on a Superdex 200 Increase 10/300 GL column, which had been preequilibrated with a running buffer (50 mM Hepes, pH 7.7, 150 mM NaCl), and then elution was carried out with the same buffer. The flow rate was maintained at 0.5 mL/min, and the absorbance of effluent was monitored at 280 nm and analyzed using Unicorn 6.3 (Cytiva).

## ELISA

A 96-well ELISA plate was coated with Fbg or citrullinated Fbg by incubating their PBS solutions (100 μL, 0.8 μg/mL) overnight at 4 °C. After washing and blocking the wells with 10% FBS-PBS, various concentrations (100 μL, 10–2000 ng/mL) of recombinant SAA (Peprotech, Cranbury, NJ) were added to the wells. The plate was incubated for 1 h at 37 °C and washed five times with PBS containing 0.5% Tween 20 (PBS-T). Then, anti-hSAA1 (1:1000, ab687, Abcam) was added to each well and further incubation was carried out for 1 h at 37 °C. The plate was washed with PBS-T five times, and anti-mouse IgG conjugated with HRP (1:1000, NA931, Cytiva) was added to each well for incubation for 1 h at 37 °C. The plate was again washed with PBS-T five times, and a TMB substrate (Thermo) was added to each well and incubated for 30 min at RT. Hydrochloric acid (3 M) was used as a quenching solution. Optical absorbance was recorded at 450 nm and 650 nm on a plate reader (SpectraMax iD5, Molecular Devices) with SoftMaxPro 7.1 (Molecular Devices).

## Fbg decomposition by plasmin

The purified Fbg and citrullinated Fbg were diluted to give a final concentration of 1 mg/mL in PBS (20 μL). Plasmin (34 units/μL, Fujifilm Wako) was diluted in PBS, and the diluted plasmin solution was added to the Fbg and citrullinated Fbg solutions. The reaction mixtures were incubated at 37 °C for 5 min to 8 h. The reactions were terminated by adding aprotinin (10,000 U/mL, Fujifilm Wako). The samples were centrifuged at 600×$g$ for 10 min. The supernatants were analyzed by SDS-PAGE and western blot.

## DLS

The DLS was measured using a Zetasizer NanoZS (Malvern Instruments, Malvern, UK). Each sample was dissolved in 50 mM Hepes, pH 7.7, and 150 mM NaCl and then was individually filtered through a membrane with a 0.22-μm pore. The samples were adjusted to a final concentration of 1 mg/mL and were prepared by mixing individual protein solutions immediately before the measurements. The correlation data were calculated using Zetasizer version 7.02 and further analyzed using the cumulant algorithm to obtain the averaged hydrodynamic diameters.

## Reverse transcription-PCR

Total RNA samples were isolated from frozen tissues using RNAiso Plus reagent (Takara Bio Inc, Shiga, Japan). Complementary DNA was synthesized with a PrimeScript II 1st Strand cDNA Synthesis Kit (Takara Bio Inc). The primers used for the PCR analyses were as follows:

(SAA1)
mSAA1-F 5′-GGATCCATGAAGCTACTCACCAGCCTGGTC-3′, mSAA1-R 5′-GGATCCTTAGTATTTGTCAGGCAGTCC-3′

(SAA2)
mSAA1-F 5′-GGATCCATGAAGCTACTCACCAGCCTGGTC-3′, mSAA2-R 5′-CTCGAGTTAGTATTTGGCAGGCAGTCCAGG-3′

(SAA3)
mSAA3-F 5′-GTCCCAGAAGGAGCTCGCAGCACG-3′, mSAA3-R 5′-CAGTAGTTGCTCCTCTTCTCGGGGG-3′

(SAA4)
mSAA4-F 5′-GGATCCATGAGGCTTGCCACCGTCATTGTC-3′, mSAA4-R 5′-CTCGAGTTAGAACTTCTCAGGAAGGCCCTC-3′

(SAAL1)
mSAAL1-F 5′-CTCGAGATGGATCGAAACCCGTCTCCTCC-3′, mSAAL1-R 5′-CTCGAGTTAAGTCTGTGCCTTCACACTTGGGAAG-3′

(EF-1α: an endogenous control)
EF-1α-F 5′-CCAATGGAAGCAGCTGGCTT-3′, EF-1α-R 5′-TCTGAGCTTTCTGGGCAGAC-3′

Genotyping primers
mSAA1-F 5′-TCTGTGGTAGGAGCTGGAAAGCATG-3′, mSAA1-R 5′-CCGAGCATGGAAGTATTTGTCTGAG-3′

mSAA2-F 5′-TCTGTGGTAGGAGCTGGAAAGCTCA-3′, mSAA2-R 5′-CCGAGCATGGAAGTATTTGTCTCCA-3′

mSAA3-F 5′-TTTCTCCCATTGCTTTGTGCTAGGC-3′, mSAA3-R 5′-AAGCTCTCTCTGAAATGGTCCAGGG-3′

mSAA4-F 5′-GTTTGAGATGGGGCCTCACTATGTAG-3′, mSAA4-R 5′-GCATGTGCTCAGAGGAAGTCAGTT-3′

mSAAL1-F 5′-GTGAGATTCCGAACAGACTCAGAGT-3′, mSAAL1-R 5′-GTTCCACCTATAGGGTTGCAGACC −3′

## Statistical analysis

Data are represented as mean ± standard error of mean (SEM). Statistical evaluation was determined using a two-tailed unpaired Student $t$-test or a one-way ANOVA with Bonferroni's test for multiple comparisons. These calculations were executed using Prism v8 software (GraphPad Software, San Diego, CA, USA). $P$ values <0.05 were considered statistically significant.

## Reporting summary

Further information on research design is available in the Nature Portfolio Reporting Summary linked to this article.

## Data availability

Microarray data were deposited in the NCBI Gene Expression Omnibus as GSE211528 (Fbg-positive/negative in pulmonary vessels of lungs from noncancer and cancer patients, Supplementary Table 1). Source data are provided with this paper.

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

## Acknowledgements
This study was supported by the Japan Society for the Promotion of Science (JSPS) KAKENHI Grant-in-Aid for Scientific Research (B) 22H02900 (S.H.) and by the Aiba Works Medical Research Grant. Y. Han was supported by Rotary Yoneyama Scholarship. Some illustrations in the figures were taken from TogoTV (©2016 DBCLS TogoTV/CC-BY-4.0).

## Author contributions
Y.Han., T.T., and S.H. designed the study and analysis. Experiments were performed by Y.Han., T.T., M.K., N.A., Y.H., H.M., W.W., H.H., Y.I., S.T., and S.H. Data analysis was performed by Y.Han., H.M., H.K., J.N., N.O., and S.H. This manuscript was prepared by S.H. with input from all authors.

## Competing interests
The authors declare no competing interests.
