## [Peer Review File · Nature Communications]

Reviewer comments, first round

Reviewer #1 (Remarks to the Author):

Yibing et al (2022) Citrullinated fibrinogen-SAAs complex causes vascular metastagenesis
The manuscript by Yibing et al describes the fascinating finding that citrullinated fibrinogen localizes to metastatic lesions and promotes their formation via an interaction with serum amyloid A proteins. In particular they show that serum amyloid A, S100A8 and HMGB1 are increased in pulmonary vessels with fibrinogen depositions, with SAA showing the highest increase. They went on to generate an SAA knockout model and demonstrated that SAA was important for lung metastasis and the citrullinated fibrinogen was detected in tumor bearing mouse lungs but not in tumor bearing SAA knockout mouse lungs. They further showed that injection of citrullinated fibrinogen promoted metastasis. Overall the results described in this manuscript are exciting and should be of general interest in the PAD field. Therefore, this reviewer recommends publication after the authors address the relatively minor comments outlined below.

1. The authors should note that targeted deletion of PAD4 in neutrophils or pharmacologic inhibition of PAD4 with JBI-589 reduced both primary tumor growth and lung metastases and substantially enhanced the effect of immune checkpoint inhibitors. (PMID: 36069973) Citation of this work would strengthen their argument for the localized deposition of citrullinated fibrinogen in the promotion of metastatic lesion formation.
2. At times, the writing could be clearer and this reviewer recommends that the manuscript be edited by a professional service.

Reviewer #2 (Remarks to the Author):

Employing human tissue analysis and murine experimental metastasis models, Yibing et al describe the role of citrullinated fibrinogen-SAAs complex in metastasis initiation in the lung. The authors provide evidence that the presence of CitFbg-SAAs complex mark metastatic hotspots and intervention with a CitFbg peptide reduced the seeding of tumor cells. While I applaud the authors for generating new tools, there are several technical and scientific caveats that need to be addressed to further advance the manuscript.

1. The authors mention that the excision of 70 kb mSAAs had no effect on viability, maturation, and fertility. Did the authors investigate the maturation of the immune system? Additionally, how is the immune cell activation affected upon the replacement of mSAAs with huSAAs? Are huSAAs functioning as acute-phase proteins in these transgenic mice? It would be crucial to understand the baseline changes in the immune surveillance following an alteration in endogenous SAAs expression.
2. Most of the experiments are conducted 24 or 48h post intravenous injection of tumor cells. What about long-term metastatic tumor formation in transgenic mice or mice administered with CitFbg or antibodies?
3. The authors talk about premetastatic niche formation. What about employing a proper spontaneous metastasis model, something like E0771 or LLC?
4. How do authors envisage comparing PAD4 targeting versus CitFbg intervention? Why don't directly manipulate PAD4 to alter the citrullination of Fbg?
5. The authors need to elaborate further on the mechanism - What leads to increased citrullination

- of fibrinogen upon tumor challenge? Is the expression level or enzymatic activity of PAD increased? Is it related to myeloid cell hotspots?
6. How can fibrinogen be citrullinated? There seem two mechanisms for citrullination; intracellular and extracellular. If it's intracellular, it's difficult to speculate that CD45+ cells citrullinated fibrinogen deposited from/around neighboring cells. It should be cell-autonomous. If it's extracellular, it's difficult again to speculate that the citrullinated fibrinogen defines the premetastatic niche while fibrinogen is usually everywhere in systemic blood circulation.
 7. Are CtiFbg-SAA hotspots essentially sites of myeloid cell infiltration? It will be interesting to investigate what causes the sporadic expression of SAAs in the lung vasculature. Clearly, not tumor cells. Further, how about the localization of myeloid cells and Fbg?
 8. In general, the quality of analytical data is not up to the mark. E.g. Western blot data shown in Fig. 1h are impossible to comprehend. Similarly, IF images are often of poor resolution where you can barely see the marked tumor cell (e.g. Fig. 2h (2) – tumor cell is not visible).
 9. Although Figs. 1g,h are key figures for this manuscript, the quality is too low to draw any conclusion. How can the authors compare the amount of fibrinogen and citrulline immunoprecipitated with SAA3 when the amount of SAA3 itself is drastically different between non-tumor-bearing and LLC-bearing samples?
 10. The authors often mention "tumor-bearing mouse lung" in the text. Are these metastatic lungs? I think the authors instead want to say lungs from the primary tumor-bearing mice. Please clarify.
 11. In Fig. S2 did the authors seed an equal number of sorted myeloid and endothelial cells, the images showed a varying number of cells. They should include a bright field or DAPI image to verify equal seeding of cells.
 12. While the authors claim that tumor cells accumulate in the vicinity of CitFbg-SAAs hotspots, fig. S2g clearly shows a tumor cell far away from Cit-Fbg in a WT lung. It is just an example but the IF data across the manuscript are often contradicting the claim made by the authors.
 13. In human tissue staining, it would be informative if the authors could mark the tumor area to distinguish the peritumoral from the core tumor area.
 14. Concerning data shown in Fig 5c, did authors inject human cells in immunocompetent mice? There is no mention in the methods section. Please clarify. Additionally, how do the authors explain the discrepant results between MCF7 (significant) and MDA-MB-231 (non-significant data) models?
 15. The authors should be careful with their interpretation of data. E.g. they mention "hSAA1 binds to hFbg and hCitFbg with comparable affinities". This sentence is in consideration when the Kd is half for hSAA1+hFbg as compared to hSAA1+hCitFbg (Fig. 6a).

Reviewer #3 (Remarks to the Author):

The manuscript by Yibing et al shows roles for citrullination in post-translational modification of fibrinogen and effects on its binding with serum amyloid A proteins, causing recruitment of cancer cells and effects on metastasis. The findings assess CitFbg deposition, identifying it as a metastatic risk, and show that CitFbg peptide can act to inhibit metastasis.

Overall, the data is convincing and supports the reported findings, but there are some significant improvements needed.

Please revise the following:

Overall read of the manuscript feels superficial with lack of in depth information in many places. The whole manuscript, particularly introduction and discussion, need to be in more depth. Results need

to be rewritten in more depth and specific and in many places the text of the results belongs in the introduction. The Methods section may need some attention for reorganising the sequence of methods described for more logical flow and more detailed information on some experiments is needed.

Specific points:

1. Introduction line 10: "It is fascinating for cancer patients..." - I doubt that "fascinating" is the correct word choice here? This sentence needs restructuring for correct delivery of the intended message.
2. In the introduction the concept of post-translational deimination/citrullination should be explained as this is the focus of the paper. Hence parts of the results section do need to move to the introduction as otherwise the paper does not have a logical flow.
3. The results should include data on PAD isoform detection that could be related to the resulting citrullination.
4. Discussion needs attention to wording, for example "This work revealed that the citrullination of fibrinogen participated in the pre-metastatic character of the soil..." Replace by "niche"?
5. Methods Human Samples: please provide a table for the samples under investigation for more details on patients and the various cancer samples. Same for number of RA samples. There is limited data given here.
6. Provide more information on the CitFbg injected (how was this prepared, which sites are citrullinated etc - this information is not clear for experiments throughout). How was the 1h timepoint selected, were other timepoints tested too? Please provide information accordingly on CitFbg for other experiments too. Possibly expand the Methods on fibrinogen purification and citrullination for more detail and move this up before other experiment description for logical flow. There is no explanation why PAD2 was used to citrullinate fibrinogen - this relates to the lack of information provided in the introduction on citrullination and the role of the different PAD isoforms. Figure 1: please ensure that figure C is bright enough, and possibly zoom in further for higher magnification. Consider adding loading controls for blots in f-h; figure j needs more contrast. Figure 2c - consider adding loading control for WB; fig g could possibly been shown at higher magnification as the labelling for SAA3 and Tumour and citrullination is hard to see. Also consider contrast. Figure 3h may also need enhancement or higher magnification for clarity of staining. Figure 5d - again some of the cell staining is not very clear, consider improving the images, also they are not all aligned or at the same size, so please check formatting.

REVIEWER COMMENTS

Reviewer #1 (Remarks to the Author):

Yibing et al (2022) Citrullinated fibrinogen-SAAs complex causes vascular metastagenesis

The manuscript by Yibing et al describes the fascinating finding that citrullinated fibrinogen localizes to metastatic lesions and promotes their formation via an interaction with serum amyloid A proteins. In particular, they show that serum amyloid A, S100A8 and HMGB1 are increased in pulmonary vessels with fibrinogen depositions, with SAA showing the highest increase. They went on to generate an SAA knockout model and demonstrated that SAA was important for lung metastasis and the citrullinated fibrinogen was detected in tumor bearing mouse lungs but not in tumor bearing SAA knockout mouse lungs. They further showed that injection of citrullinated fibrinogen promoted metastasis. Overall, the results described in this manuscript are exciting and should be of general interest in the PAD field. Therefore, this reviewer recommends publication after the authors address the relatively minor comments outlined below.

#1. The authors should note that targeted deletion of PAD4 in neutrophils or pharmacologic inhibition of PAD4 with JBI-589 reduced both primary tumor growth and lung metastases and substantially enhanced the effect of immune checkpoint inhibitors. (PMID: 36069973) citation of this work would strengthen their argument for the localized deposition of citrullinated fibrinogen in the promotion of metastatic lesion formation.

Thank you for your information and suggestion. We added this paper in the reference section (ref 43), and discussion about PAD functions in endothelial cells and myeloid cells in pre-metastatic lungs. We also added data to confirm PAD2- and PAD4-mediated citrullination in those cells using siRNA. This point was also raised by other reviewers.

#2. At times, the writing could be clearer, and this reviewer recommends that the manuscript be edited by a professional service.

We rewrote the revised manuscript, and it was edited by a professional service before the resubmission.

Reviewer #2 (Remarks to the Author):

Employing human tissue analysis and murine experimental metastasis models, Yibing et al describe the role of citrullinated fibrinogen-SAAs complex in metastasis initiation in the lung. The authors provide evidence that the presence of CitFbg-SAAs complex mark metastatic hotspots and intervention with a CitFbg peptide reduced the seeding of tumor cells. While I applaud the authors for generating new tools, there are several technical and scientific caveats that need to be addressed to further advance the manuscript.

Thank you for the reviewer's constructive comments. We have addressed the reviewer's comments and suggestions as below.

#1. The authors mention that the excision of 70 kb mSAAs had no effect on viability, maturation, and fertility. Did the authors investigate the maturation of the immune system? Additionally, how is the immune cell activation affected upon the replacement of mSAAs with huSAAs? Are huSAAs functioning as acute-phase proteins in these transgenic mice? It would be crucial to understand the baseline changes in the immune surveillance following an alteration in endogenous SAAs expression.

To survey the immune system in wild-type, mSAAs^{-/-} and hSAAs/mSAAs^{-/-} mice, we analyzed T cells, B cells, NK cells, dendritic cells, and CD11b⁺ myeloid cells in the peripheral blood taken from these mice, by using flow cytometry. To check the immune responses driven by tumor-derived instigations, these mice were tested with or without tumor-conditioned medium application as an acute stimulation in tumor progression. There was no apparent difference in the immune cell populations among these mice. The results data were summarized in supplementary Fig. S1h. We further confirmed that lungs derived from primary tumor-bearing wild-type (mSAA^{+/+}) and hSAAs/mSAAs^{-/-} mice contained almost the same number of CD11b⁺ cells. Because these cells are recruited to the lungs as a result of primary tumor-derived immune responses, these data indicate that these mice equip similar immune cells. We added this data in the revised manuscript as supplementary Fig. S5a.

#2. Most of the experiments are conducted 24 or 48h post intravenous injection of tumor cells. What about long-term metastatic tumor formation in transgenic mice or mice administered with CitFbg or antibodies?

We added two data to show long-term (over 3 weeks) metastatic tumor formations after tumor cell injection. First, we checked that CitFbg-administered wild-type mice had larger number of macroscopic metastases than control (PBS-injected) mice. Second, we showed that hSAAs promoted lung metastasis 3 weeks after an injection of human cancer cells such as MDAMB231 or MCF7 in hSAAs/mSAAs^{-/-} and mSAAs^{-/-} mice with Rag1^{-/-} background. These data were included in Figs. 3c and 5g, respectively.

#3. The authors talk about premetastatic niche formation. What about employing a proper spontaneous metastasis model, something like E0771 or LLC?

We showed data using 3LL, which is a spontaneous metastatic cell line of Lewis lung carcinoma (LLC), in supplementary Fig. S1j (first submission Fig. S1i) and added statements in the revised text and legend.

#4. How do authors envisage comparing PAD4 targeting versus CitFbg intervention? Why don't directly manipulate PAD4 to alter the citrullination of Fbg?

We examined PAD2- and PAD4- dependent citrullination of Fbg. Endothelial cells (ECs) and CD11b⁺ myeloid cells were isolated from lungs of primary tumor-bearing mice, and siRNA for PAD2 or PAD4 was applied to these cells. As shown in the data presented in Fig. 2e, knockdown of PAD2 and PAD4 suppressed citrullination of Fbg in the cases of lung ECs. It was also observed that PAD4 knockdown in lung CD11b⁺ myeloid cell reduced citrullination of Fbg. The PAD2- and PAD4- mediated citrullination was also supported by the experiments for comment #5.

#5. The authors need to elaborate further on the mechanism - What leads to increased citrullination of fibrinogen upon tumor challenge? Is the expression level or enzymatic activity of PAD increased? Is it related to myeloid cell hotspots?

We searched factors that would stimulate PAD-protein expression using our lung organ culture system. Among candidates such as mSAA3, hSAA1, S100A8, CCL2, SDF1 and VEGF, each of them has been reported as a pre-metastatic inducer, mSAA3 significantly induced PAD2 and PAD4 in the mouse lungs. Recombinant hSAA1 was also effective to increase PAD2 expression. We presented these data in the revised manuscript as supplementary Fig. S2f. Regarding myeloid cell hotspots, please see our response to comment #7.

#6. How can fibrinogen be citrullinated? There seem two mechanisms for citrullination: intracellular and extracellular. If it's intracellular, it's difficult to speculate that CD45⁺ cells citrullinated fibrinogen deposited from/around neighboring cells. It should be cell autonomous. If it's extracellular, it's difficult again to speculate that the citrullinated fibrinogen defines the premetastatic niche while fibrinogen is usually everywhere in systemic blood circulation.

Because we clarified that SAAs induced PAD-dependent citrullination as stated in the response to comment#5, we decided to check whether CitFbg-SAA aggregation could be established in culture medium with human lung ECs. Human lung ECs were cultured in the presence of purified Fbg or CitFbg with or without recombinant SAA1 for one week, and SAA1 and citrullinated proteins suspended in the culture media were detected by immunostainings. This method allows us to observe protein complexes in the medium but not in the cells. Our results shown in Fig. 6d revealed that EC incubation with CitFbg and SAA1 gave more aggregations than other conditions. Thus, it is highly likely that SAAs expressed on ECs meet CitFbg, which was citrullinated by PAD-proteins derived from ECs in the extracellular sphere, to form an aggregation complex.

We newly carried out *in vitro* aggregation assays. SAA1, CitFbg, and/or SAA1-CitFbg were incubated at 37°C for 1 day to 1 week, and resulting complexes were analyzed by DLS. These DLS data imply that SAA “seeding”, a macromolecular complex to become a core, accelerated formation of CitFbg aggregations. The data presented in supplemental Fig. S6c exhibits that addition of CitFbg after SAA1 solo-incubation generated larger complexes than SAA1 and CitFbg co-incubation. To form large size CitFbg aggregation, the aggregation complex must deprive CitFbg from

the vicinity. This indicates that formation of large aggregation reduces CitFbg concentration. This is why we were able to observe limited number of CitFbg aggregations in our mouse model system. Because we needed to obtain aggregation of CitFbg-SAA in culture medium without containing cell components, we could not carry out similar experiment using floating CD11b⁺ myeloid cells, which might function as a PAD4 supplier. We showed the mechanism model in Supplementary Fig. S7d. Also, we added PAD-proteins from ECs and myeloid cells as a generator of CitFbg in the mechanism model (Fig. 7f).

#7. Are CitFbg-SAA hotspots essentially sites of myeloid cell infiltration? It will be interesting to investigate what causes the sporadic expression of SAAs in the lung vasculature. Clearly, not tumor cells. Further, how about the localization of myeloid cells and Fbg?

We measured distances between SAA-CitFbg signals and CD11b⁺ myeloid cells in our frozen lung sections. Our conclusion is that there seemed to be no correlation between SAA-CitFbg signals and CD11b⁺ myeloid cells. We showed this data in supplementary Fig. S2k and added a sentence in the revised manuscript to specify the involvement of CD11b⁺ myeloid cells. Because CD11b⁺ myeloid cells can freely move in the lung tissue, they do not have to stay near SAA-CitFbg even if they have a functional relationship with SAA-CitFbg. It is possible that CD11b⁺ myeloid cells were involved in the SAA-CitFbg formation process as a source of PAD-proteins. Thus, we included CD11b⁺ myeloid cells in the hotspot formations as shown in Fig. 7f and supplementary Fig. S7d.

#8. In general, the quality of analytical data is not up to the mark. E.g. Western blot data shown in Fig. 1h are impossible to comprehend. Similarly, IF images are often of poor resolution where you can barely see the marked tumor cell (e.g. Fig. 2h (2) – tumor cell is not visible).

We rerun western blots using the same samples used for the first submission or newly prepared samples. In the revised Fig. 1h, bands near 90 kDa can be observed in WT but not SAAs^{-/-} samples. We replaced IF images so that readers can recognize

tumor cells clearly. Regarding original Fig. 2h, new images showing tumor cells (in the mirror section) more clearly than the initial version were presented instead, as Fig. 2i.

#9. Although Figs. 1g, h are key figures for this manuscript, the quality is too low to draw any conclusion. How can the authors compare the amount of fibrinogen and citrulline immunoprecipitated with SAA3 when the amount of SAA3 itself is drastically different between non-tumor-bearing and LLC-bearing samples?

We recognize that our explanations about Figs. 1g and 1h are very confusing. The purpose of Fig. 1g is to exhibit an interaction between CitFbg and SAA3 in the mouse lung environment, but there is no intention to quantitate amount of Fbg or CitFbg bound on SAA3. To show the SAA3 specific interactions, an SAA3-IP sample prepared from no tumor-bearing mouse lungs was a good control because it has only trace amount of SAA3. In Fig. 1g, the fact that the SAA3-IP sample from no tumor-bearing lungs did not show bands specific to CitFbg indicates that CitFbg did not bind anti-SAA3-protein G beads because there was no SAA3. In other words, CitFbg did not show non-specific binding to the antibody bound beads. Finally, Fbg and citrullinated protein bands observed in the lanes of tumor-bearing lung sample support the CitFbg-SAA3 interaction.

As stated in the response to comment#8, Fig. 1h was replaced with better images. We added GAPDH blots for input samples (small part of samples used for the IP was loaded on a different gel) as control to show lung lysates contained substantial amount of lung proteins. This point was also raised by reviewer 3.

We added several sentences for the detailed explanation in the revised text.

We first detected a faint ~90 kDa band of fibrinogen in lung samples obtained from tumor-bearing and TCM-stimulated mice, whereas there was no such counterpart signal in no tumor-bearing mouse lung samples (Fig. 1f). The 90 kDa band as well as three fibrinogen subunits were clearly detected in the immunoprecipitation product of lung lysate from tumor-bearing condition by anti-SAA3 antibody (Fig. 1f, right lane). This data implied that there was an SAA3-fibrinogen interaction with the 90 kDa component. We again confirmed the upshifted fibrinogen band in the SAA3-binding precipitate using lung lysate derived from tumor-bearing mice. The lung lysate obtained from the no tumor-bearing mice contained undetectable levels of SAA3, so that anti-SAA3

pull-down produced no SAA3 band (Fig. 1g, left). This SAA3-negative lane did not show bands unique to fibrinogen, suggesting that the upshifted fibrinogen band could not be nonspecifically detected.

We then found that this upshift was due to citrullination modification (Fig. 1g right). We further showed that citrullinated fibrinogen was increased in lungs from primary tumor-bearing- wild-type compared to SAAs^{-/-} mice (Fig. 1h). We improved Western blot data using two anti-citrullination antibodies with control Fbg blotting (revised Fig. 1h). We also added GAPDH blot as sample control (revised Figs. 1f, g and h). Based on these data, we showed three results in this part. First, primary tumors induced SAAs in the lungs. Second, SAA3 bound Fbg and CitFbg in lungs in the primary tumor-bearing condition. Third, citrullination of Fbg was induced in an SAAs-dependent manner.

#10. The authors often mention “tumor-bearing mouse lung” in the text. Are these metastatic lungs? I think the authors instead want to say lungs from the primary tumor-bearing mice. Please clarify.

Thank you for this suggestion. We first described the term “tumor-bearing mouse lung” as no metastatic lung derived from the primary tumor-bearing mouse in the revised text.

We use the term “tumor-bearing mouse lungs” to specify “lungs from primary tumor-bearing mice with no metastasis”. The “tumor-bearing mouse lungs” also indicates that lungs are in the pre-metastatic phase. (Page 5)

#11. In Fig. S2 did the authors seed an equal number of sorted myeloid and endothelial cells, the images showed a varying number of cells. They should include a bright field or DAPI image to verify equal seeding of cells.

We replaced image that CD144⁺ ECs from SAA^{-/-} tumor-bearing mice on

fibrinogen-coated dish with DAPI after staining by anti-citrullination-antibody. The CD11b⁺ and F4/80⁺ cells were washed away during the staining process because they are not adherent cells. We saw remaining a few immune cells in the immunostained images (supplementary Fig. S2e in the revised manuscript).

#12. While the authors claim that tumor cells accumulate in the vicinity of CitFbg-SAAs hotspots, fig. S2g clearly shows a tumor cell far away from Cit-Fbg in a WT lung. It is just an example but the IF data across the manuscript are often contradicting the claim made by the authors.

We carefully checked all the images including Fig S2g. We replaced Fig S2g new one for better understanding. (new Fig. S2h)

#13. In human tissue staining, it would informative if the authors could mark the tumor area to distinguish the peritumoral from the core tumor area.

We first checked metastatic tumor area in the lungs using HE staining (For the reviewer Data). Then, we prepared paraffin blocks of metastasis-free lung tissues close to the metastatic area. Because our choice contained no-tumor area only, we are unable to show tumor and peritumoral area in the paraffin blocks.

#14. Concerning data shown in Fig 5c, did authors inject human cells in immunocompetent mice? There is no mention in the methods section. Please clarify. Additionally, how do the authors explain the discrepant results between MCF7 (significant) and MDA-MB-231 (non-significant data) models?

We injected human tumor cells into C57BL/6 background in 24 h to examine attachment ability of human cancer cell to the hCitFbg-deposition site in mouse vascular system. As reviewer pointed out, there was not significant difference between PBS and CitFbg data in the MDAMB231 experiments. We repeated the experiments to

find out the statistical significance (p value = 0.0223, Fig. 5c). Thus, we found that the PBS vs CitFbg difference of MDAMB231 was not as large as that of MCF7, but both were statistically significant ($p < 0.05$).

To demonstrate the importance of hCitFbg in the premetastatic environment, we decided to use mice with Rag1^{-/-} background. These mice allow us to track down the effects of hCitFbg-SAA and human cancer cells in the long-term metastasis assay. In this assay, we excised lungs from each mouse 3 weeks after the human tumor cell injection. The assay data revealed that CitFbg-pretreated hSAA/mSAA^{-/-}/Rag1^{-/-} mice had more lung metastasis than CitFbg-pretreated mSAA^{-/-}/Rag1^{-/-} mice. We added these results in the revised manuscript as Fig. 5g and supplementary Fig. S5g. To avoid confusion, we noted mouse background on Fig. 5a and 5g, because Rag1^{-/-} background mice were used for the long-term metastasis assay only.

#15. The authors should be careful with their interpretation of data. E.g., they mention “hSAA1 binds to hFbg and hCitFbg with comparable affinities”. This sentence is in consideration when the K_d is half for hSAA1+hFbg as compared to hSAA1+hCitFbg (Fig. 6a).

It is true that K_d value of hSAA1-hCitFbg was half for that of hSAA1-hFbg. If both hSAA1-Fbg and hSAA1-CitFbg bindings are 1:1 binding, the conclusion would be that hSAA1 binds to hFbg rather than hCitFbg. At the same time, however, B_{max} value of hSAA1-hCitFbg was twice as much as that of hSAA1-hFbg. This indicates that CitFbg binds more SAA1 molecules than Fbg. In this case, simple comparison of K_d values between hSAA1-Fbg and hSAA1-CitFbg is not enough to draw any conclusion. Our ELISA data showed that there was no significant difference in SAA1 binding between on Fbg and on CitFbg in the region of low SAA1 concentration. We realized our initial submission was confusing because we used apparent K_d values to discuss the interactions between hSAA1 and Fbg or CitFbg. We rewrote this part.

Data for reviewer #2

Metastatic site

We prepared paraffin blocks of metastasis-free lung tissues (Block 1) close to the metastatic tumor (Block 2). Because we examined no-tumor area only, we are unable to show tumor and peri-tumoral area in the same slide.

Reviewer #3 (Remarks to the Author):

The manuscript by Yibing et al shows roles for citrullination in post-translational modification of fibrinogen and effects on its binding with serum amyloid A proteins, causing recruitment of cancer cells and effects on metastasis. The findings assess CitFbg deposition, identifying it as a metastatic risk, and show that CitFbg peptide can act to inhibit metastasis.

Overall, the data is convincing and supports the reported findings, but there are some significant improvements needed.

Please revise the following:

Overall read of the manuscript feels superficial with lack of in depth information in many places. The whole manuscript, particularly introduction and discussion, need to be in more depth. Results need to be rewritten in more depth and specific and in many places the text of the results belongs in the introduction. The Methods section may need some attention for reorganising the sequence of methods described for more logical flow and more detailed information on some experiments is needed.

Specific points:

#1. Introduction line 10: "It is fascinating for cancer patients..." - I doubt that "fascinating" is the correct word choice here? This sentence needs restructuring for correct delivery of the intended message.

Thank you for your comment. Some of our word choices were not appropriate for patients. We modified the sentence.

This research would be very beneficial for cancer patients, as this would alert them in the very early phases, known as the pre-metastatic phase, which would help to prevent initial metastasis and occlude the secondary metastasis, because of which, their prognosis would be far better than those requiring remedying of established metastases.

#2. In the introduction the concept of post-translational deimination/citrullination should be explained as this is the focus of the paper. Hence parts of the results section do need to move to the introduction as otherwise the paper does not have a logical flow.

We rebuilt the introduction section as suggested by the reviewer to make a logical flow of this paper more articulate.

#3. The results should include data on PAD isoform detection that could be related to the resulting citrullination.

The revised manuscript contains the PAD isoform detection in lung endothelial cells and CD11b⁺ myeloid cells. Data is presented in Fig. 2e. We examined PAD2- and PAD4- dependent citrullination of Fbg. Endothelial cells (ECs) and CD11b⁺ myeloid cells were isolated from lungs of primary tumor-bearing mice, and siRNA for PAD2 or PAD4 was applied to these cells. As shown in the data presented in Fig. 2e, knockdown of PAD2 and PAD4 suppressed citrullination of Fbg in the cases of lung ECs. It was also noteworthy that PAD4 knockdown in lung CD11b⁺ myeloid cell reduced citrullination of Fbg.

#4. Discussion needs attention to wording, for example "This work revealed that the citrullination of fibrinogen participated in the pre-metastatic character of the soil..." Replace by "niche"?

We replaced “soil” by “niche” in the sentence and carefully checked Discussion section in the revised text.

#5. Methods Human Samples: please provide a table for the samples under investigation for more details on patients and the various cancer samples. Same for number of RA samples. There is limited data given here.

We added human sample information in the revised manuscript as supplementary Table 2. We prepared paraffin sections from two RA and non-RA patients, based on the diagnosis. We added number in the methods section.

#6. Provide more information on the CitFbg injected (how was this prepared, which sites are citrullinated etc - this information is not clear for experiments throughout). How was the 1h timepoint selected, were other timepoints tested too? Please provide information accordingly on CitFbg for other experiments too. Possibly expand the Methods on fibrinogen purification and citrullination for more detail and move this up before other experiment description for logical flow. There is no explanation why PAD2 was used to citrullinate fibrinogen - this relates to the lack of information provided in the introduction on citrullination and the role of the different PAD isoforms.

The reason why 1 h timepoint was selected in this study came from our *in vitro* experiment. To prepare Fbg- or CitFbg coating dishes, we usually take 1 h incubation prior to use. These dishes were tested by immunostaining to confirm the presence of Fbg or CitFbg. Our *in vitro* cell attachment assay demonstrated that 1 h incubation of MCF7 and MDAMB231 cells allowed cells to adhere on the CitFbg-coated dish as presented in Fig. 7c. Moreover, it is necessary to avoid secondary effects caused by the injection of exogenous materials. Generally, mice receiving a protein injection show inflammatory responses at earliest a few hours later, and these inflammatory responses mask what we would like to observe. Preparation of CitFbg and related information were described in the method section.

We increased description of Fbg and CitFbg preparations and moved up to the top of method section. Previous study conducted by another research group demonstrated that human Fbg was citrullinated by PAD2 and PAD4. Their further analysis found that 22 arginine residues in A α chain were citrullinated by PAD2 but some of them were not citrullinated by PAD4. In addition, there was no citrullination unique to PAD4. Our biological assessment to determine the deimination enzyme revealed that at least PAD2 secreted from ECs was participating in Fbg citrullination in the extracellular sphere, although there is no doubt about importance of PAD4. Together, we decided to use PAD2 to prepare CitFbg.

#Figure 1: please ensure that figure C is bright enough, and possibly zoom in further for higher magnification. Consider adding loading controls for blots in f-h; figure j needs more contrast. Figure 2c - consider adding loading control for WB; fig g could possibly be shown at higher magnification as the labelling for SAA3 and Tumour and

citrullination is hard to see. Also consider contrast. Figure 3h may also need enhancement or higher magnification for clarity of staining. Figure 5d - again some of the cell staining is not very clear, consider improving the images, also they are not all aligned or at the same size, so please check formatting.

We modified figure presentations as suggested by the reviewer. Also, we added loading control for western blots by running the same samples, prepared for the originally submitted data set, in a different gel.

Reviewer comments, second round

Reviewer #1 (Remarks to the Author):

The authors have addressed my concerns and have appear to have addressed the concerns of the other reviewers. I recommend acceptance of the paper.

Reviewer #2 (Remarks to the Author):

The manuscript has much improved and can further advance with the following comments.

1. Fig. S1j, what do the authors mean by metastatic nodules of size 0mm? Perhaps, it will be clearer to include the number of lung metastases per mouse as well. Alternatively, the total metastatic area per lung area can be shown.
2. Fig. S1i, the authors should include statistics when comparing CD45+ staining for substantiating their conclusions in lines 152-153.
3. Similar to comment 2, the authors draw major conclusions from Fig. S2a in lines 190-192, without even doing a statistical comparison of their data. Here, it is trivial that there are more myeloid cells in the lungs of tumor-bearing mice as compared to the lungs of non-tumor-bearing mice. A more important question is whether the deletion of SAA in SAA^{-/-} mice affects myeloid cell infiltration. Here again, authors refrain from a direct statistical comparison between tumor-bearing WT and SAA^{-/-} mice. Authors should reanalyze their statistics before concluding.
4. Lines 192-193, the authors quote, "The citrullination signals around CD11b+, F4/80+, and endothelial cells were prominently observed in the tumor-bearing mouse lungs (Fig. 2a and Fig. S2b)". Similarly, here the authors fail to show statistical data to substantiate their conclusions. Deletion of SAA does not affect citrullination signals while comparing tumor-bearing WT or SAA^{-/-} mice.
5. Given the broad readership of Nature Communications, the authors must take extra caution and explain the rationale behind their experiments and perhaps have some more empathy for the reader. This reviewer recommends professional English editing of the manuscript. Even after extensive editing by the authors, the manuscript in its current version is still too complex to read, and oftentimes, the flow of experiments is just not logical and clear enough.
6. The label of the Y-axis is missing in Fig. 2d.

Reviewer #3 (Remarks to the Author):

The authors have addressed my comments raised in the previous review round.

REVIEWER COMMENTS

Reviewer #2 (Remarks to the Author):

#1. Fig. S1j, what do the authors mean by metastatic nodules of size 0mm? Perhaps, it will be clearer to include the number of lung metastases per mouse as well. Alternatively, the total metastatic area per lung area can be shown.

Thank you for your comment. In the submitted version, Fig S1j represents metastatic nodule size and number. The figure showed that WT mice had two 6-mm nodules, one 5-mm nodule, and six 2-mm nodules. We recognized this rendering was very confusing. We added data and reanalyzed metastatic nodule numbers for WT and SAAs^{-/-}. The revised Fig. S1j simply shows number of lung metastasis, which makes our data understandable.

#2. Fig. S1i, the authors should include statistics when comparing CD45⁺ staining for substantiating their conclusions in lines 152-153.

Thank you for your comment. We added CD45 staining data to recalculate the statistical significance. The results were reflected in the revised Fig. S1i.

#3. Similar to comment 2, the authors draw major conclusions from Fig. S2a in lines 190-192, without even doing a statistical comparison of their data. Here, it is trivial that there are more myeloid cells in the lungs of tumor-bearing mice as compared to the lungs of non-tumor-bearing mice. A more important question is whether the deletion of SAA in SAA^{-/-} mice affects myeloid cell infiltration. Here again, authors refrain from a direct statistical comparison between tumor-bearing WT and SAA^{-/-} mice. Authors should reanalyze their statistics before concluding.

Thank you for your comment. We newly prepared tumor-bearing mouse lungs. These frozen sections were used for CD11b staining to increase the data points for Fig. S2a. Our revised result exhibits an increase of CD11b positive cells in tumor-bearing mouse lungs compared to those in the lungs of no tumor-bearing mouse and tumor-bearing SAAs^{-/-} mouse with statistical significance. $P < 0.0001$ is shown in WT no tumor-bearing vs. WT

tumor-bearing, and $P = 0.0023$ is shown in WT tumor-bearing vs. SAAs^{-/-} tumor-bearing. We added these data points and p-values for the statistical comparisons in Fig. S2a.

#4. Lines 192-193, the authors quote, “The citrullination signals around CD11b⁺, F4/80⁺, and endothelial cells were prominently observed in the tumor-bearing mouse lungs (Fig. 2a and Fig. S2b)”. Similarly, here the authors fail to show statistical data to substantiate their conclusions. Deletion of SAA does not affect citrullination signals while comparing tumor-bearing WT or SAA^{-/-} mice.

Thank you for your comment. We understand that Fig. 2a was somewhat misleading. Fig. 2a data showed citrullination levels on the surface of EC, CD11b⁺, and F4/80⁺ cells. Although cell surface citrullination correlates with ability to citrullinate the substrate to some extent, it is not enough to describe the precise activity of citrullination, because these data disregarded effects of secreted proteins. In fact, as shown in Fig. S2k, some citrullinated fibrinogen were very close to CD11b⁺ cells but vast majority were away from them. Given the fact that CD11b⁺ cells were moving agilely in the tissue, it was highly likely that PAD enzymes were functioning when secreted from the CD11b⁺ cells. Thus, we concluded that we should present PAD2 and PAD4 data instead. PADs detections can describe the ability of citrullination more accurately. To analyze PAD2 and PAD4 immunostaining data, we decided to prepare new samples. Because we keep limited number of animals, our focus was on the comparison of no tumor bearing with tumor-bearing. We analyzed many lung sections (total 1,700 sections) including newly prepared samples and Fig. 2a and Fig. S2b were replaced by new graphs. Analysis of PAD4 levels in endothelial cells and CD11b⁺ cells indicated that PAD4 was upregulated in WT tumor-bearing lungs compared with WT no tumor-bearing lungs and SAAs^{-/-} tumor bearing lungs. Statistical analyses show that WT no tumor-bearing vs. WT tumor-bearing ($p = 0.014$, PAD4-EC and $p = 0.019$, PAD4-CD11b) and WT tumor-bearing vs. SAA^{-/-} tumor-bearing ($p = 0.0016$, PAD4-EC and $p = 0.004$, PAD4-CD11b) were statistically significant. These data are interpreted that PAD4 enzyme levels were increased in tumor-bearing mouse lungs in an SAAs dependent manner. Fig. 2c data support this conclusion. Our in vitro assay shown in Fig. 2c demonstrated that WT CD144⁺ and CD11b⁺ cells have much higher citrullination ability than SAAs^{-/-} counterparts. PAD2 data are shown in Fig. S2b. PAD2 data did not show statistical significance but data pattern (upregulated in WT tumor-bearing EC and CD11b) was similar to that of PAD4.

#5. Given the broad readership of Nature Communications, the authors must take extra caution and explain the rationale behind their experiments and perhaps have some more empathy for the reader. This reviewer recommends professional English editing of the manuscript. Even after extensive editing by the authors, the manuscript in its current version is still too complex to read, and oftentimes, the flow of experiments is just not logical and clear enough.

Thank you for your comment. As abovementioned in the responses to comments #3 and #4, we renewed the data to make the entire logical flow more articulate. Statements of this part in this manuscript was rewritten and professional English editing helped the improvement. We added the certificate issued by the third party.

#6. The label of the Y-axis is missing in Fig. 2d.

Thank you for your comment. We added the label of the Y-axis in Fig. 2d.

Reviewer comments, third round

Reviewer #2 (Remarks to the Author):

I warmly congratulate the authors for making the effort to advance the manuscript.